

# Seismic Background Noise Levels in Italian Strong Motion Network

Simone Francesco Fornasari[1], Deniz Ertuncay[1], and Giovanni Costa[1]

[1]SeisRaM Working Group, Department of Mathematics and Geosciences, University of Trieste, Via Eduardo Weiss 4, 34128 Trieste, Italy

**Correspondence:** Simone Francesco Fornasari (simonefrancesco.fornasari@phd.units.it)

**Abstract.** Italian strong motion network monitors the seismic activity of Italy and its surrounding with more than 700 stations. Thanks to the upgrade of the stations with continuous data acquisition, it is possible to measure the noise level of the strong motion network. In this study, we used the recorded background noise to estimate the variations in the noise levels of the network. Data recorded in 2019 and 1st of January to 30th of April 2022 are used to understand the noise level of the stations

and data from the COVID-19 lockdown period are used to see the effect of the anthropogenic sources on the background noise. To do that, power spectrum density is calculated for the continuous stations. It is found that more than half of the stations exceed the background noise model designed for strong motion stations by Cauzzi and Clinton (2013) in at least one of the calculated periods. Considering the characteristics of the instrumentation at the stations and their deployment often near urban areas, we focused on relatively short periods (≤5s), interested by anthropic noises. Stations can be noisier during the day, up

to 14 decibels and during the weekday, up to 5 decibels in short periods. Noise level differences between day - night decrease with an increasing period as the human-related high - frequency effects of humans are attenuated. As expected, the noisiest stations are located in densely populated areas such as center of Naples, whereas the quietest stations are located far away from cities. The swell, sea, and wind effects, on the other hand, are not observed in stations. During the COVID-19 lockdown, noise levels dropped to 6.5 decibels in daytime and 12.5 decibels on weekdays. Noise levels are reduced by around 2 decibels in

0.1s, in which cultural noise is predominant. Furthermore, we found that the vehicles have measurable effects on noise levels.

## 1 Introduction

Seismic stations record the vibration of the ground that is given by the superposition of multiple sources. The definition of seismic noise varies based on the target of each specific study. Since most of the seismic networks are created to detect seismic events, earthquakes, volcanic activities, quarry blasts, nuclear explosions and so on, all other vibrations are considered as

(ambient) noise. On the other hand, noise can be used for the characterization of layers of the earth (Shapiro et al., 2005). Noises can also be sub-categorized based on their source such as; i) seismic recorder (Ringler and Hutt, 2010), ii) temperature changes (Doody et al., 2018), iii) ocean & sea waves (Webb, 1998; McNamara and Buland, 2004; Bonnefoy-Claudet et al., 2006; Cauzzi and Clinton, 2013; D'Alessandro et al., 2021; Anthony et al., 2021), iv) wind (Mucciarelli et al., 2005; Bonnefoy-Claudet et al., 2006; D'Alessandro et al., 2021; Anthony et al., 2021), v) anthropic activities (McNamara and Buland, 2004;

Bonnefoy-Claudet et al., 2006; Cauzzi and Clinton, 2013; Vassallo et al., 2019; D'Alessandro et al., 2021; Anthony et al., 2021) (Figure 1).



The level of noise affects the quality of the seismic signals, hence the ability to detect seismic events. To be able to monitor the seismic sources, seismic networks need to have knowledge about the noise content of the networks. To characterize the noise at a given station, the frequency content of the noise is calculated via power spectrum density (PSD). The above-mentioned

noise sources can be seen in different frequency bands of the PSD (D'Alessandro et al., 2021). Various models are created to interpret the noise levels. The model of Peterson (1993) is widely used to define the lower and upper bounds of the recorded noise as a baseline. In that study, low (New Low Noise Model, NLNM) and high (New High Noise Model, NHNM) levels of noise models are developed using a worldwide catalogue from a wide variety of seismic stations. Cauzzi and Clinton (2013) developed the accelerometer low-noise (ALNM) and high-noise (AHNM) models using accelerometric data from the Swiss

Seismological Service (Clinton et al., 2011) and very broad-band along with accelerometric data from Southern California Seismic Network (California Institute of Technology and United States Geological Survey Pasadena, 1926). The ALNM is computed as the lower boundary of 5-percentile PSD amplitudes observed on rock sites while the AHNM is computed as a particular combination of accelerometric sensors with a given gain and response with dataloggers. In ALNM, instrument and data logger noise are dominant at all frequencies, whereas in AHNM, urban noise, microseismic activities, and data logger

systems dominate the short periods, mid-range periods, and long periods, respectively. This model is widely used as the baseline model for strong motion sensors (Ringler et al., 2015, 2020).

To reduce the recorded noise, seismic stations should be installed away from anthropogenic noises such as roads, major cities, factories etc. However, not all seismic stations are placed in quiet locations since other factors weigh in the selection of the "optimal" locations; even though human activity increases the noise level, the installation in buildings and infrastructure

provides information about potential damages occurring during seismic events. Italian Civil Protection Department (DPC) operates the integrated Italian Strong Motion Network (RAN, Rete Accelerometrica Nazionale in Italian, Gorini et al. 2010; Costa et al. 2022) to monitor strong motions at a national level. RAN is the result of cooperation between the Italian government, regions, and local authorities that has been carried out for more than 25 years.

In this paper, we focus on the noise in RAN accelerometric network. To do that, we analyze the data coming from 528

continuous stations between 2019 and 2022. To see the effect of the COVID-19 pandemic, we compare the 2019 and 2022 noise data with the one from the Italian nationwide lockdown in March 2020. Starting from the end of 2021, a vast amount of RAN stations are changed with new generation recorders. Furthermore, they have been converted from triggered to continuous recording, enabling us to study the nationwide noise levels of the accelerometric network for the first time thanks to continuous data. In Section 2, we explain the properties of the RAN network and the time coverage of the data. In Section 3 section,

the data preprocessing and PSD calculation features are explained. Background noise levels are presented in Section 4 and the possible noise sources are discussed in Section 5. During the covid lockdown the opportunity is raised to see the noise level changes due to human activity and how 'silent' the stations can be. Variations in the noise levels during the COVID-19 and non-COVID-19 time periods, along with several noise sources that we can clearly identify, are interpreted, and the overall background noise of the RAN network is presented in Section 6. For simplicity COVID-19 lockdown period is called lockdown

period and non-COVID-19 time period is called no-lockdown.




## 2 Data

The integrated National Accelerometric Network (RAN, Costa et al. 2022) comprises more than 700 stations managed by cooperating Italian governmental bodies and regional and local authorities. The network is the combination of 3 accelerometric networks namely, the RAN, owned and managed by the DPC (Gorini et al., 2010; Zambonelli et al., 2011; Costa et al., 2022); the Friuli Venezia Giulia Accelerometric Network (RAF, Rete Accelerometrica Friuli Venezia Giulia in Italian, Costa et al. 2010) in the North-East, owned and managed by the University of Trieste (UniTS); and the Irpinia Seismic Network (ISNet, Weber et al. 2007) in the South, owned and managed by Analysis and Monitoring of Environmental Risk society (AMRA). However, some of them are triggered stations and it is not possible to calculate noise levels. Hence we have 528 stations that provided continuous data in the time range that we are interested in. RAN stations have generally a standardized installation near urban areas (see Table 1) in free-field conditions, with instruments placed on an isolated pillar anchored on rock or put inside of the sediments. Since 2020, the network transitioned from triggered to continuous recording thanks to improvements/updates in the recording instrumentation.

Data from 2019 are used to characterise background noise information from the RAN network along with seasonal, daily, and hourly changes. Data collected during the 2020 COVID-19 lockdown (9 March - 4 May 2020) provide information about the noise levels when the anthropogenic sources nationwide were minimal. Data from 2022 (1 January - 30 April 2022) are used to study the post-lockdown noise level and, thanks to the great increase in the number of continuous stations, provide better coverage of the Italian territory.

Thereinafter the combined data from 2019 and 2022 will be referred to as non-lockdown data, as opposed to the data from 9 March - 4 May 2020 which will be addressed as lockdown data. The location and data availability for each station is presented in Figure 2.

## 3 Method

The method introduced by McNamara and Buland (2004) represents the de facto standard for the evaluation of PSDs. This method was originally developed as a tool for monitoring the status of seismic stations: as such, the original parameters used for the computation of the PSDs and the use of smoothing and averaging provide a way to reduce the storage and computation costs involved, but can be limiting when the method is extended to scientific uses, as shown by Anthony et al. (2020).

The method implemented to compute the PSDs partially mirrors the one by Anthony et al. (2021), which in turn is an adaptation of McNamara and Buland (2004). Each daily recording is divided into $90\,\mathrm{min}$ windows with $50\,\%$ overlap, each one subsequently divided into $15\,\mathrm{min}$ subwindows with $75\,\%$ overlap: as pointed out by Anthony et al. (2020), the window length becomes less relevant for higher frequencies and noisier stations, which are the conditions of the present study. Data completeness above $90\,\%$ is required for each $90\,\mathrm{min}$ window. During preprocessing, the data are linearly detrended, the gaps are linearly interpolated, and a Hann window is applied to limit spectral leakage (Peterson, 1993; Anthony et al., 2021). No binning and smoothing are performed during the PSDs computation. Similarly to Anthony et al. (2021), we performed a one-third octave average over the PSDs: the averaging bandwidth can be assumed as a reasonable trade-off between the obtained





spectral resolution and the accuracy in the broadband noise sources characterization in each band. The parameters used for the
evaluation of the PSDs in our study, along with the ones used in McNamara and Buland (2004); D'Alessandro et al. (2021);
Anthony et al. (2021) are reported in Table 2.

To study specific patterns in the noise levels over time, the PSDs are studied by grouping them over different time ranges.
To study the effects of anthropic noise it is a common practice to consider the variations between day (08:00 - 18:00) and night
(20:00 - 07:00) and between weekday (Monday - Friday) and weekend (Saturday - Sunday). Similarly, the variations between
summer and winter are analysed to check seasonal variations of the noise levels. To study seasonal variations we limited
our analysis to 2019 being the only currently available year-long dataset with continuous recordings unaffected by lockdown
measures. The statistics related to these variations are computed over the daily difference of the medians of each group.

## 4   Results

The results obtained applying the method introduced in the Method are shown in Figure 3 for a few selected stations and for
some periods of interest, namely $0.1\,\mathrm{s}$, $0.25\,\mathrm{s}$, $0.5\,\mathrm{s}$, $1\,\mathrm{s}$, $2\,\mathrm{s}$, and $5\,\mathrm{s}$ (Figure 4).

The period-wise median of the PSDs for each station is computed and interpreted as the representative noise level. Since
the RAN network is a strong motion network, we are mainly interested in periods less than $5\,\mathrm{s}$. Anthropogenic sources can
have a major role in the noise content of short periods (Table 1) which also provide vital information for seismic parameters
estimation, seismic engineering and building monitoring. The quietest and noisiest RAN stations for each period of interest are
reported in Table S1 with the related noise level and the station placement (in the Supplement each station is described along
with an explanation about the possible noise source in the nearby area).

The RAN network has relatively high noise levels in high frequencies. Numerous stations exceed the levels defined by
Cauzzi and Clinton (2013). The median noise at each station, presented in Figure 4, and the AHNM have been compared and
the results are reported in Table 3. $1\,\mathrm{s}$ is the period for which we have the highest rate of excedence with $34.4\,\%$ of the stations
exceeding the AHNM level. The probability density function calculated over the median PSD of all stations can be seen in
Figure 5. The median values are always below the AHNM model for the frequency range of interest. Between $0.1\,\mathrm{s}$ and $2\,\mathrm{s}$,
stations located in the Po valley and the area from Ischia Island to Naples have relatively high noise levels. Stations around
Ischia Island have the same trend in higher periods.

Under the common assumption that the anthropic noise decreases during the night hours and during the weekend, we
characterised the contribution of human activities to the ambient noise levels. At $98.2\,\%$ of the stations, nighttime has lower
noise levels on average with respect to the daytime (Figure 6).

Another common assumption is that the noise levels are lower during the weekends due to the reduction in working activities.
Considering the data from 2019 and 2022 are used we studied the changes in the noise levels between weekdays and weekends
(Figure 7).





We have analyzed the seasonal variation of very long period noises, as shown in Figure 8. The results show that winters are noisier than the summers as suggested by Anthony et al. (2021); D'Alessandro et al. (2021), however there is no significant variation among long periods even though main noise sources for each period are different (Figure 1).

### 4.1   COVID-19 Lockdown

In the early periods of the pandemic, Italy introduced a full lockdown in the country, which limited the daily activity of the
general public. Lockdown is started on the 9th of March 2020 (8th of March in Northern Italy) and ended on the 4th of May 2020. After the nationwide lockdown new measures were set region-wise to decrease the spread of the virus which is harder to track since the measures have changed over time. Because of that, we only analyze the full lockdown between March and May 2020. To do that, results from the 2019 and 2022 are compared with the lockdown period.

To observe the noise level changes during the lockdown, 309 stations that were continuously recording during 2019 and/or
2022 and the lockdown period are selected. We presented the noise differences in periods of $0.1\,\mathrm{s}$, $0.25\,\mathrm{s}$, $0.5\,\mathrm{s}$, $1.0\,\mathrm{s}$, $2.0\,\mathrm{s}$, and $5.0\,\mathrm{s}$ for the common stations (Figure 9). Furthermore, hour and day - specific results are also presented in daytime - nighttime (Figure 10). Weekday-weekend differences are also calculated but not presented in the paper for the sake of simplicity. The figures can be seen in Supplement Figure S2.

### 5   Discussion

In Table 1, it is shown that most of the stations are located in urban areas, in which the cultural noise ($\leq 1\,\mathrm{s}$) is increased due to human activity. Even though there are several stations (e.g., DST2) that are located in the settlement zones, they are slightly far away from the densely populated areas. Nevertheless, the source of the high frequency noises can be linked to the human activity in most cases (see Section 5.2).

The interpretation of the background noise in Italian strong motion network can be done in three different ranges that are
low periods ($<1\,\mathrm{s}$), medium range periods ($\geq 1\,\mathrm{s}$, $\leq 5\,\mathrm{s}$), and long periods, ($>5\,\mathrm{s}$). As mentioned before, in the lower periods, human activities are the main source of the background noise. In 273 stations of 525 noise levels exceed the AHNM developed by Cauzzi and Clinton 2013 (Table 3).

The effect of the human activity on noise levels can be seen by comparing daytime noise to nighttime noise, for which the human activity is assumed to be reduced. As seen in Figure 6, the majority of the stations are noisier during the day for
periods less than $1\,\mathrm{s}$. The noise difference between day and night decreases with increasing periods, but the nationwide trend of days being noisier is valid for $0.1\,\mathrm{s}$, $0.25\,\mathrm{s}$, and $0.5\,\mathrm{s}$. The same pattern can be seen in broadband stations located in Italy (D'Alessandro et al., 2021).

In the weekday - weekend variations, the same trend can be followed in short periods. Figure 7 shows that weekdays of were noisier with respect to weekends in almost all stations. Depending on the settlement's characteristics, noise level change can
have large or small values on the weekend. If a station is located in a settlement where on the weekend human related noises



are not changing (e.g. touristic areas), the power change between weekday and weekend will be small. The same logic applies to the stations located on the outskirts of the settlements, since the high-frequency noises attenuate rapidly with distance.

In the medium range periods, there are multiple noise sources that have been identified by previous studies (Figure 1). Cauzzi and Clinton (2013) stretches the cultural noise up to $3\,\text{s}$ whereas D'Alessandro et al. (2021) indicates that wind and swell related noises are dominant between $1\,\text{s}$ to $10\,\text{s}$. Consequently, variations in the noise sources in $2\,\text{s}$ and $5\,\text{s}$ can be found by analyzing the daily, weekly and seasonal changes.

Day and night differences follow the trend that is seen in shorter periods in most of the network. However, in $1\,\text{s}$ the day and night differences are nulled in central Italy, whereas stations located at Po valley, Ischia island, and Naples continue to be noisier during the day. The majority of the stations exceed the AHNM threshold in $1\,\text{s}$, and the noise levels do not change during the night, which means that the anthropogenic effects are not the source. Even though in $2\,\text{s}$ and $5\,\text{s}$ there is a general trend of having higher noise levels during the day time, the power change is very small. We believe that the effects of sea, swell, and/or wind at our stations have not been identified and thus, do not have a significant role on the noise levels. There is no clear correlation between the noise level at our stations and their distance to the coastline, as also shown in Figure 4.

On the weekdays and weekends, stations start being noisier on weekends with decreasing power change. In the Po valley, the general trend of a high noise level diminishes starting from $2\,\text{s}$ in average and in the same periods, unlike the day and night difference, weekends follow the same trend.

In long periods, the effects of wind, swell, sea, pressure, and instrumental noises are in action (Figure 1). The difference between the noise levels in the winter and summer periods of 2019 can be seen in Figure 8. There is almost no change in terms of noise level changes within periods. Moreover, there is no change in noise levels from shores to inland and from high altitudes (Alps, Apenines) to low altitudes (Po valley). We believe that the main source of the noise in long periods is the instrumental noise, as indicated by Cauzzi and Clinton (2013), in which accelerometric stations are used, as in our study.

In the study of D'Alessandro et al. (2021), it is stated that in periods between $0.83\,\text{s}$ and $8.33\,\text{s}$ noises are higher in coastal areas with respect to the inland. However, in our study, there is no evidence of such a pattern. In Figure 4, there are some areas that follow the pattern found by D'Alessandro et al. (2021), such as in Naples, noise levels are higher than in the stations that are East of Naples inland. In $1\,\text{s}$ only the stations in Naples are in agreement with D'Alessandro et al. (2021) and in our study noise levels are much higher in other parts of Italy. The same trend can be seen in longer periods ($>5\,\text{s}$). There are numerous stations located in the Po valley with high noise levels even though they are far away from the sea, and several stations located in the Alps in North West Italy. In $0.1\,\text{s}$, we have noisy stations in Po valley, Puglia, and the eastern part of Sicily, where our stations are noisier than the ones analyzed in D'Alessandro et al. (2021). However, in short periods our results are in agreement with the study of D'Alessandro et al. (2021) in other parts of Italy. Human-made activities dominate the low period periods of the noises content and high noise levels can be linked to the activities that are occurring in the area where anthropic sources are present. Reduction in human activity can be seen in Figure 6 in which almost all stations have lower noise levels at night with respect to their daytime counterparts.


## 5.1 COVID-19 Lockdown

Previous studies showed that during the COVID-19 lockdown there was decrease in noise levels due to the reduction of human-related activities, and as recorded by both broadband (Poli et al., 2020; Xiao et al., 2020; Lecocq et al., 2020; Somala, 2020; Dias et al., 2020; Roy et al., 2021; Grecu et al., 2021; Cannata et al., 2021) and strong motion (Yabe et al., 2020; Łukasz Ściśło et al., 2021) stations. These activities were affecting our stations more significantly since several of the RAN stations are located inside or near to the public buildings.

Human activity was reduced during the lockdown period of the COVID-19 pandemic, since individuals were only allowed to move up to $500\,\mathrm{m}$ diameter from their homes and only essential workers were exempt from the distance restrictions. Many public institutes worked remotely in most of their units, which also reduced the human activity in the public buildings where some of our stations are located. This leads to the reduction in noise levels in (Table 4). In the $0.1\,\mathrm{s}$, there is almost $2\,\mathrm{dB}$ noise reduction between the median noise difference between lockdown and no - lockdown time periods at the stations (Figure 9a).

The difference has the lowest reduction in $1\,\mathrm{s}$ but the noise levels are higher during this period with respect to AHNM (Figure 4). In Apennines there are numerous station in which noise levels between $0.5\,\mathrm{s}$ to $2\,\mathrm{s}$ have not been affected by the lockdown (Figure 9c-e). Being the instrument self noise the dominant source in the long periods, we limited out analysis of the effect of the lockdown to periods up to $5\,\mathrm{s}$. For all the periods considered, the lockdown period is on average quieter during daytime than the 2019 - 2022, with an average noise level reduction of $1.0\,\mathrm{dB}$.

The change between daytime and nighttime are visible especially on shorter periods ($\leq 0.5\,\mathrm{s}$, Figure 10a,c,e). Changes in the daytime are more significant than the changes in the nighttime between the lockdown and no-lockdown time span. In the shorter periods, both more stations are noisier during the daytime in the no - lockdown period and the power change is almost always greater in the daytime, with respect to the nighttime difference. All stations are noisier both during the daytime and nighttime in periods shorter than $0.5\,\mathrm{s}$, whereas, in periods of $1\,\mathrm{s}$ and $2\,\mathrm{s}$, stations in The Apennines have similar power change in both daytime and nighttime. There is a clear trend of noisier day and nights in Southern Italy in $2\,\mathrm{s}$ and it can be partially

traced in $5\,\mathrm{s}$. Even though numerous stations have relatively high noise levels in $2\,\mathrm{s}$ (Figure 4e), there is no particular pattern in this period with respect to other parts of Italy.

## 5.2 Case Study: Stations located in Trieste

CARC (latitude: $45.652\,60$, longitude: $13.770\,00$) and DST2 (latitude: $45.658\,90$, longitude: $13.801\,30$) stations are part of

215 the RAF network and are located in Trieste, North-East Italy. Even though the distance between these two stations are less than $3\,\mathrm{km}$, there is a significant difference in noise levels among them. DST2 station is located at the basement of one of the Mathematics and Geosciences Department buildings of the University of Trieste that sits on a deep Flysh deposits (Figure 11). CARC station is located on the ground floor of the Palazzo Carciotti which is located in the city center of Trieste and was built in the early 19th century. It crosses with one of the major roads in the city center and the building is surrounded by multistory

residential buildings. Historically, this area was salina (Figure 12) and the area is filled with the $27\,\mathrm{m}$ depth material (Fitzko et al., 2007) to cover up the salina to expand the city in the 18th century.



In Figure 13, median of lockdown and no-lockdown dates are presented. In order to see the hourly changes in noise levels, 90 min PSDs are plotted, separately. In the lower periods ($<1$ s) where anthropogenic noises are more dominant, CARC station is noisier in both time ranges. In the daytime noise levels surpass the upper limit of Cauzzi and Clinton (2013) model, whereas

in nighttime they are close to the upper limit. During the lockdown, both stations have lower amplitudes in high frequencies. At CARC station, there is significant decrease in noise between 20:00 and midnight. Between 1 Hz and 50 Hz, median noise levels are decreased 8.68 dB and 2.36 dB for DST2 and CARC stations, respectively.

It is worth to consider that even though the noise levels are dropped in CARC station, it is not as significant as DST2. In the lockdown period human activity was limited but it was not fully halted. On the other hand, in the San Giovanni Campus of

University of Trieste where DST2 is located, almost all human activity is ceased. The university campus was already quieter than the city center and the closest residential zone is around 100 m away from the station. On top of that, the library of the department which is located on the ground floor of the building was closed during the lockdown.

### 5.3  Vehicle Noise

As mentioned before in Section 1 some of the seismic stations are positioned in public buildings that are near the main roads.

This caused relatively noisy stations and one of the major source for the noise is cars. Various schools, municipality, and governmental buildings are used as a shelter for the seismic stations and these building are tend to positioned in urban areas with convenient transportation infrastructure.

To demonstrate the effect of the cars in seismic noise measures, PLTA (latitude: 41.886 40, longitude: 14.789 30) station is chosen (Figure 14). PLTA station is located next to the municipality building of Palata in Central Italy, which has 2 intersections

50 m away from it. Cars are detected manually in 13 days of 2019 by hand. In total, 7289 car related signals with time duration between 5 s and 20 s are chosen for further analysis in which Fast Fourier transforms (FFTs) are calculated (Figure 14). In Figure 15, it can be seen that the frequency information of the cars and peaks in PSD are overlapping which means that in the period range between 9.5 s and 50 s, cars can be considered as the main source of the noise.

### 6  Conclusions

In this study, the noise levels of RAN network are analyzed. To do that, continuous stations of the network are selected for 2019 and 2022. Thanks to the modernization of the stations, for the first time noise level of the Italian strong motion can be analyzed in its full scale. Recently, noise levels of Italian broadband network is available (D'Alessandro et al., 2021) and our study completes the strong motion noise levels of the Italian territory. To make the analysis, PSDs over 90 min windows of signals are processed. It is found that a significant number of stations (up to 51.3 %) in relatively short periods

($\leq 1$ s) have higher noise levels than the AHNM that is defined for accelerometers in Switzerland by Cauzzi and Clinton (2013).

As presented in Section 4 RAN network have several very noisy stations located in the city centers. We must stress out that fundamental duty of the RAN network is to provide ground motions of the locations where civil defense may need to provide assistance in post-disaster (eg. strong earthquake) situations. Even though some of these stations are noisy (eg. CSA7), they





are well capable of providing the true nature of the ground motion if there is a strong earthquake nearby, hence they are able to
serve for their purpose (Costa et al., 2022). On the other hand, there are large number of stations with low noise levels. These
stations not only capture the amplitudes of large magnitude events but also the small ones. Hence, they can be used in a wider
portion of observational seismology related studies. The median noise levels (Figure 4) provide an overview of the background
noise of the network and they can be used as a station selection criterion depending on the nature of the future research.

Some of the stations in the RAN network are positioned inside or just outside the governmental buildings such as schools,
municipalities (528 of the stations are installed in settlements) whereas some of them are located away from settlements. As a
result of this, stations such as CARC have high noise levels. CARC also has higher noise levels than DST2 station, which is
located less than $3\,\mathrm{km}$ away from it but inside the building with relatively scarce human activity (Figure 13). Another problem
of being inside the settled areas is the vehicle noise. It is found that vehicles can dominate the background noise in short periods
(Figure 15). On the other hand, it shows that installing a station slightly away from the city center, if possible, may increase
the data quality significantly without losing much information about the area of interest. Having the example of the CARC and
DST2 stations, network operators can reconsider relocating the seismic stations away from the city center. We are well aware
of the fact that it is not always an easy task due to various factors (eg. logistics, agreements among network operators and local
authorities).

To see the daily variation of the noise levels of the station, noise levels between the hours of daytime (08:00 - 18:00) and
nighttime (20:00 - 07:00) are compared. It is found that in short periods where human - made activities dominate the seismic
records, daytime is noisier than nighttime. This trend can be seen in some stations also in longer periods, but it cannot be
generalized for all stations.

In the longer periods ($\geq 1\,\mathrm{s}$), various studies showed that wind, swell, sea, pressure, and instrumental noises are the dominant
sources. However, in our study there is no clear pattern of swell and sea effect (between $1\,\mathrm{s}$ and $40\,\mathrm{s}$) can be followed by
analyzing the results from the stations located near the seaside and away from the coastal line (Figure 4). For instance, stations
in the Alps can be noisier than the stations in Genoa, which is a city located on the coast of the Tyrrhenian Sea. There is also no
seasonal pattern seen in very long periods (Figure 8). In periods between $2\,\mathrm{s}$ to $5\,\mathrm{s}$, winter is noisier as expected from previous
studies(D'Alessandro et al., 2021); but in longer periods it is reversed and the median noise difference between winter and
summer are generally constant with increasing period. This leads us to interpret the main source of the noise in these periods
to be the instrumental noise as Cauzzi and Clinton (2013) indicated.

During the COVID-19 lockdown in Italy that is applied between March and May 2020, noise levels are reduced due to
several measures that limited the human activity. Its effect can be seen for all the considered periods with an average reduction
in the noise level of $1.0\,\mathrm{dB}$ (and up to $2.9\,\mathrm{dB}$ at $0.0625\,\mathrm{s}$) during the daytime. The effect of the lockdown also affected the
weekday and weekend variations of the noise levels.

*Code and data availability.* The analysis has been performed using the data and metadata from the Italian Strong Motion Network (RAN,
Gorini et al. 2010; Costa et al. 2022). Data and materials along with the developed models can be found in a dedicated GitHub repository.



https://doi.org/10.5194/nhess-0-1-2022-supplement

*Author contributions.* Conceptualisation, all authors.; methodology, S.F.F.; software, S.F.F.; data curation, all authors; writing—original draft preparation, D.E. and S.F.F.; writing—review and editing, all authors; visualisation, S.F.F. and D.E.; supervision, G.C.; project administration,
G.C.; funding acquisition, G.C. All authors have read and agreed to the published version of the manuscript.

*Competing interests.* The contact author has declared that none of the authors has any competing interests.

*Acknowledgements.* This study received financial support from the Italian Civil Protection - Presidency of the Council of Ministers.





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





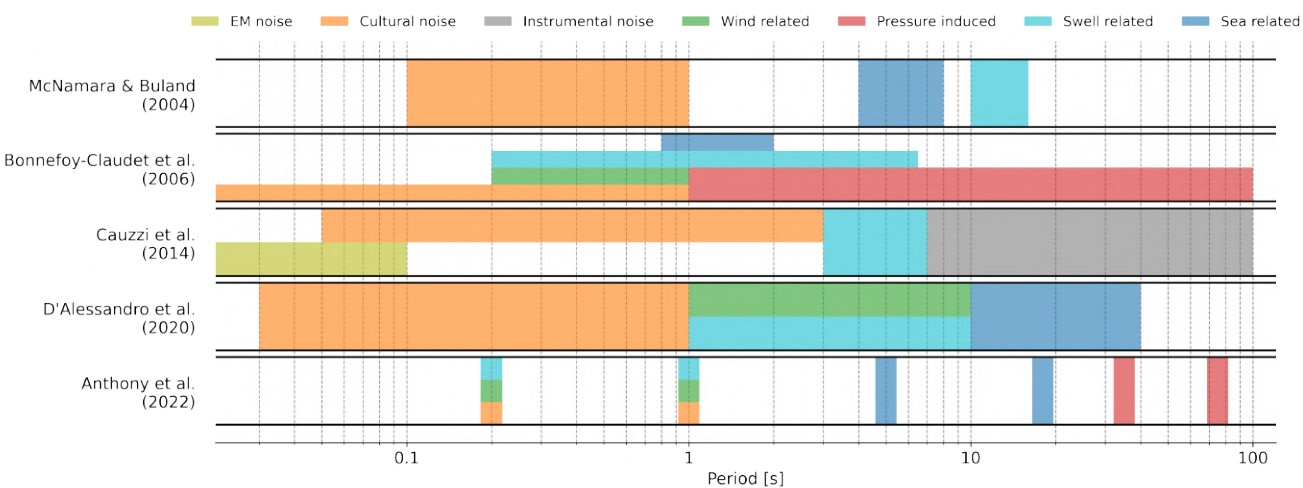

**Figure 1.** Main noise sources for different period bands from the studies of McNamara and Buland (2004); Bonnefoy-Claudet et al. (2006); Cauzzi and Clinton (2013); D'Alessandro et al. (2021); Anthony et al. (2021)

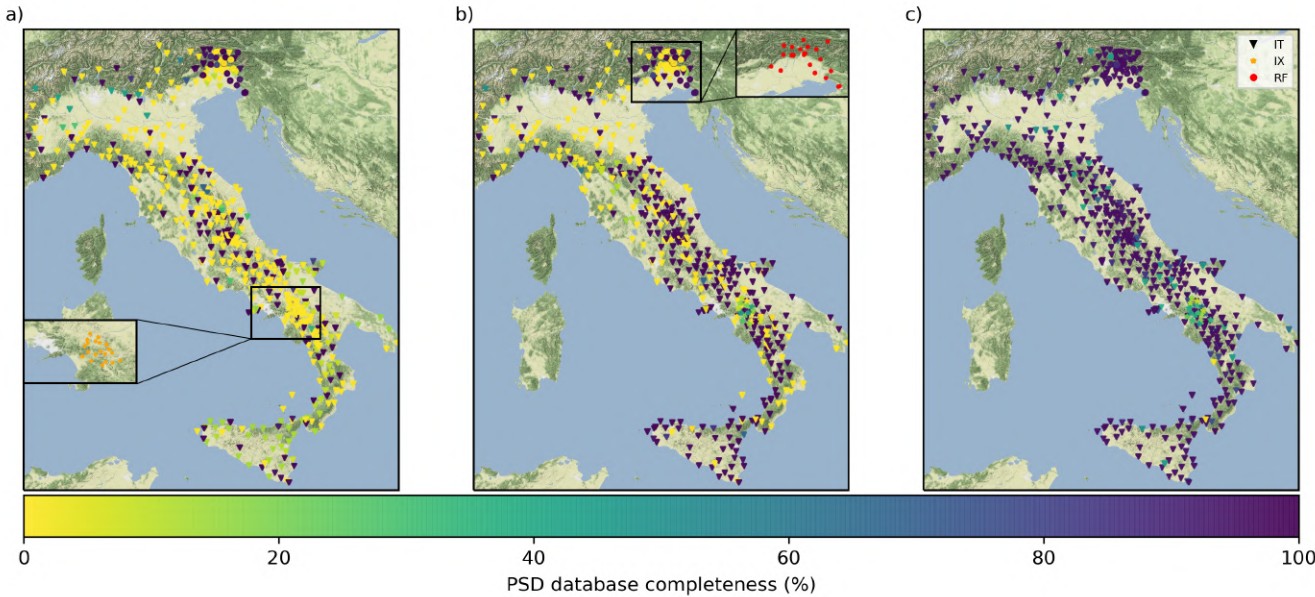

**Figure 2.** Data availability of the stations in a) 2019, b) lockdown period, and c) 2022. In a) the close up box highlights ISNet (IX) and in b) the close up box highlights RAF (RF). Basemap data are retrieved from © Stamen Design.


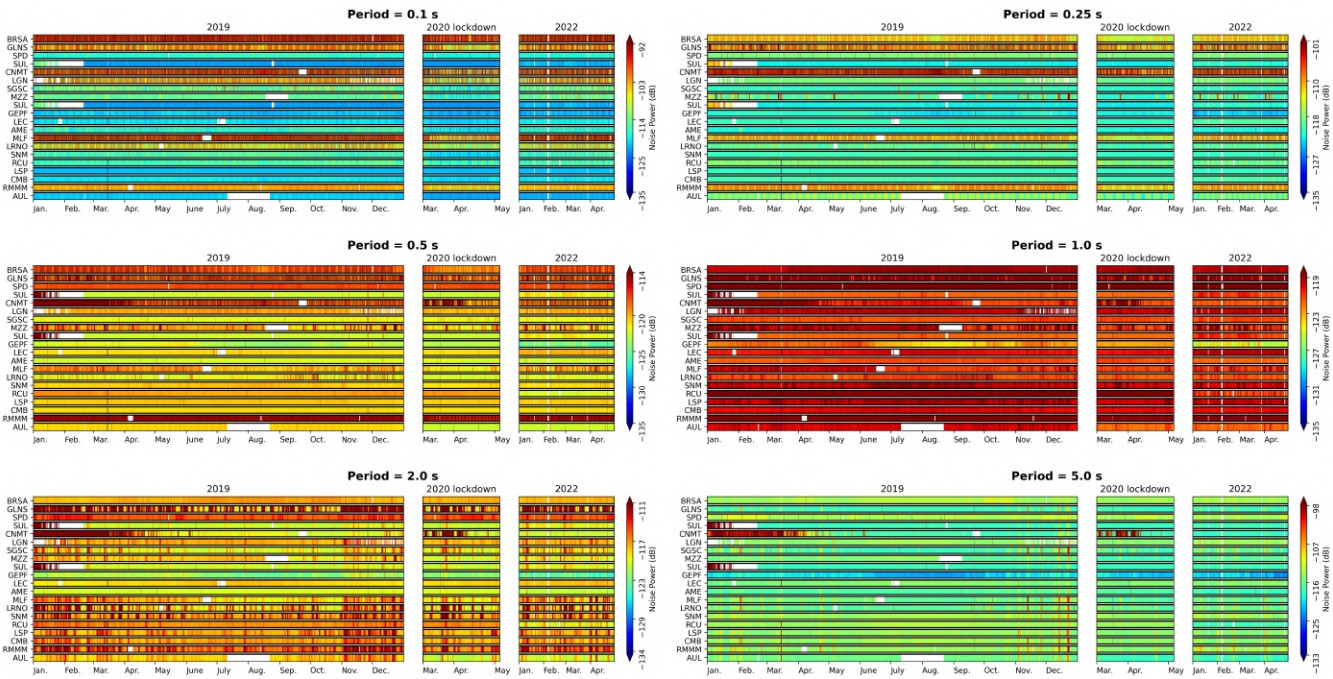

**Figure 3.** PSD timeseries for several selected stations in periods of 0.1 s, 0.25 s, 0.5 s, 1 s, 2 s, and 5 s. The limits of the color scale are based on the models by Cauzzi and Clinton (2013).

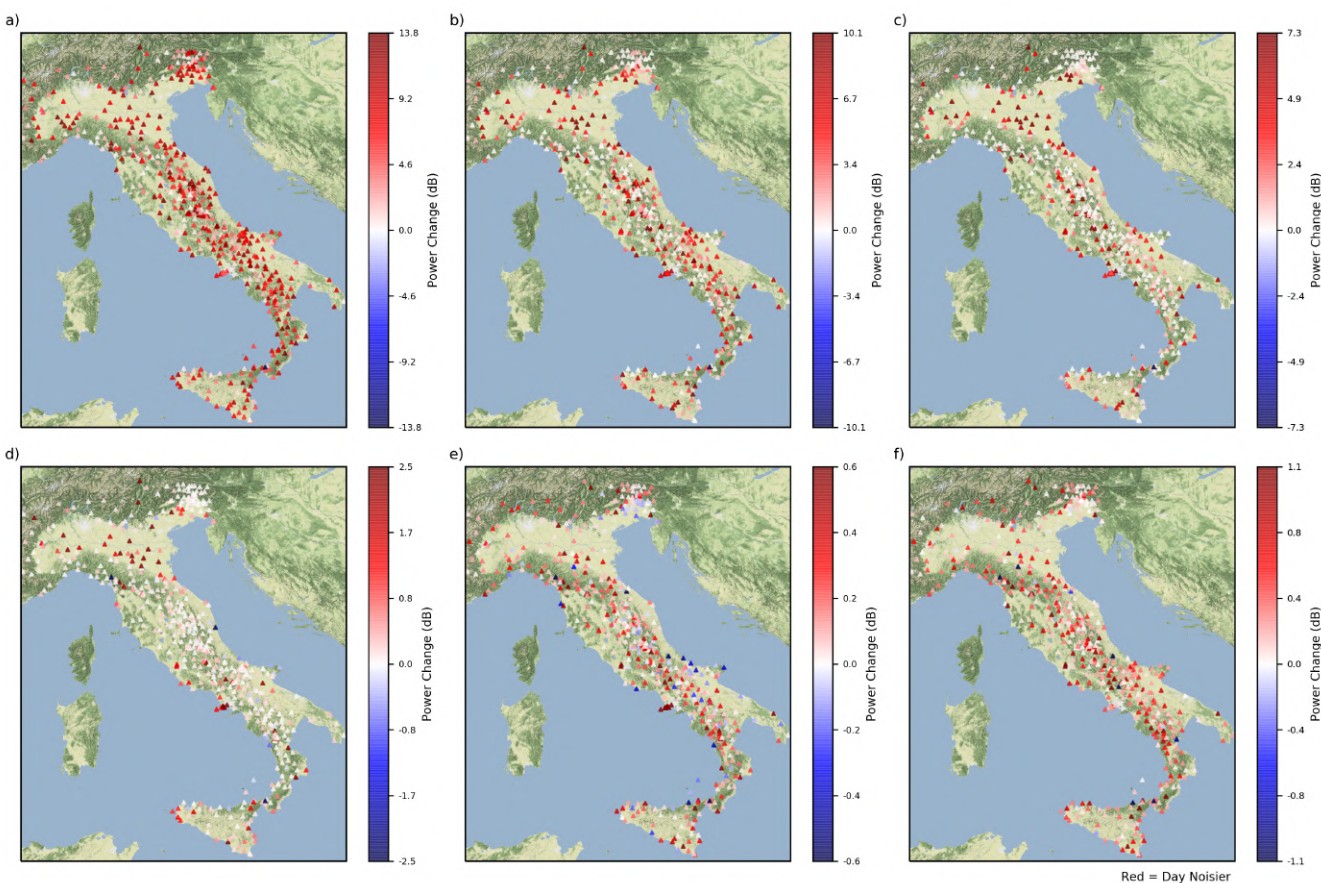

**Figure 4.** Median vertical component noise maps in one-third octave bands around a-g) 0.1 s, 0.25 s, 0.5 s, 1 s, 2 s, 5 s, 16 s, 32 s, and 80.6 s. Upper and lower limits of the color bar are defined by the model developed by Cauzzi and Clinton (2013). Vertical components are presented in the following figures and Electronic Supplement. Background noise levels of all calculated periods can be found in Figure S1. Basemap data are retrieved from © Stamen Design.

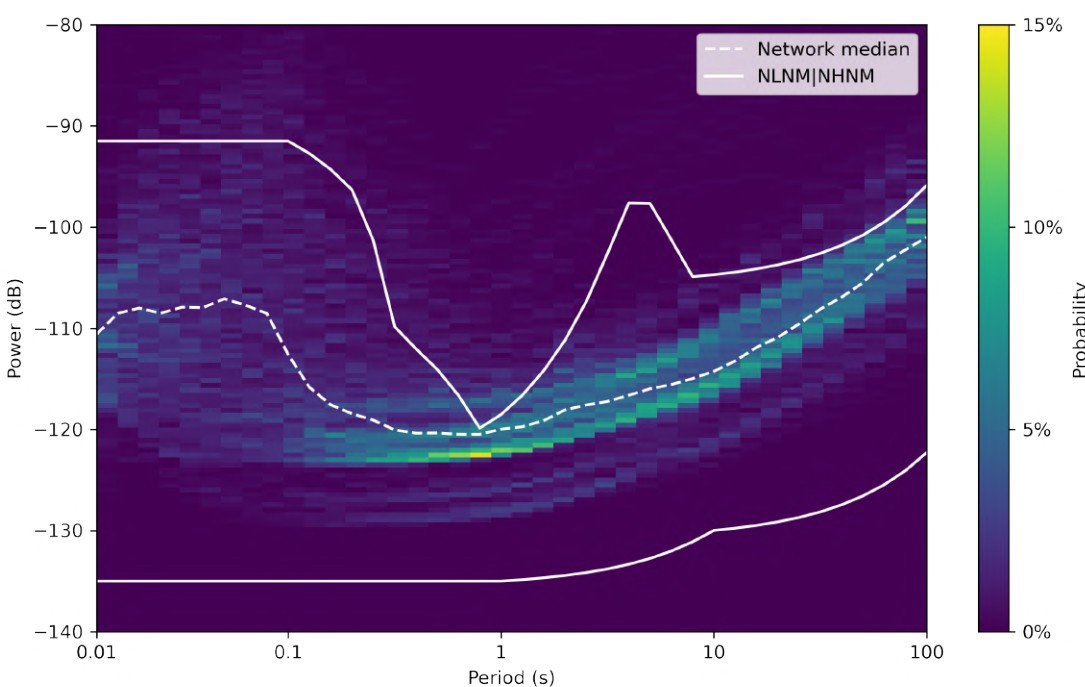

**Figure 5.** Probability density function of medians of PSDs of all stations. Dashed white line represent the median of the network and solid white lines represent NLNM and NHNM defined by Cauzzi and Clinton (2013).

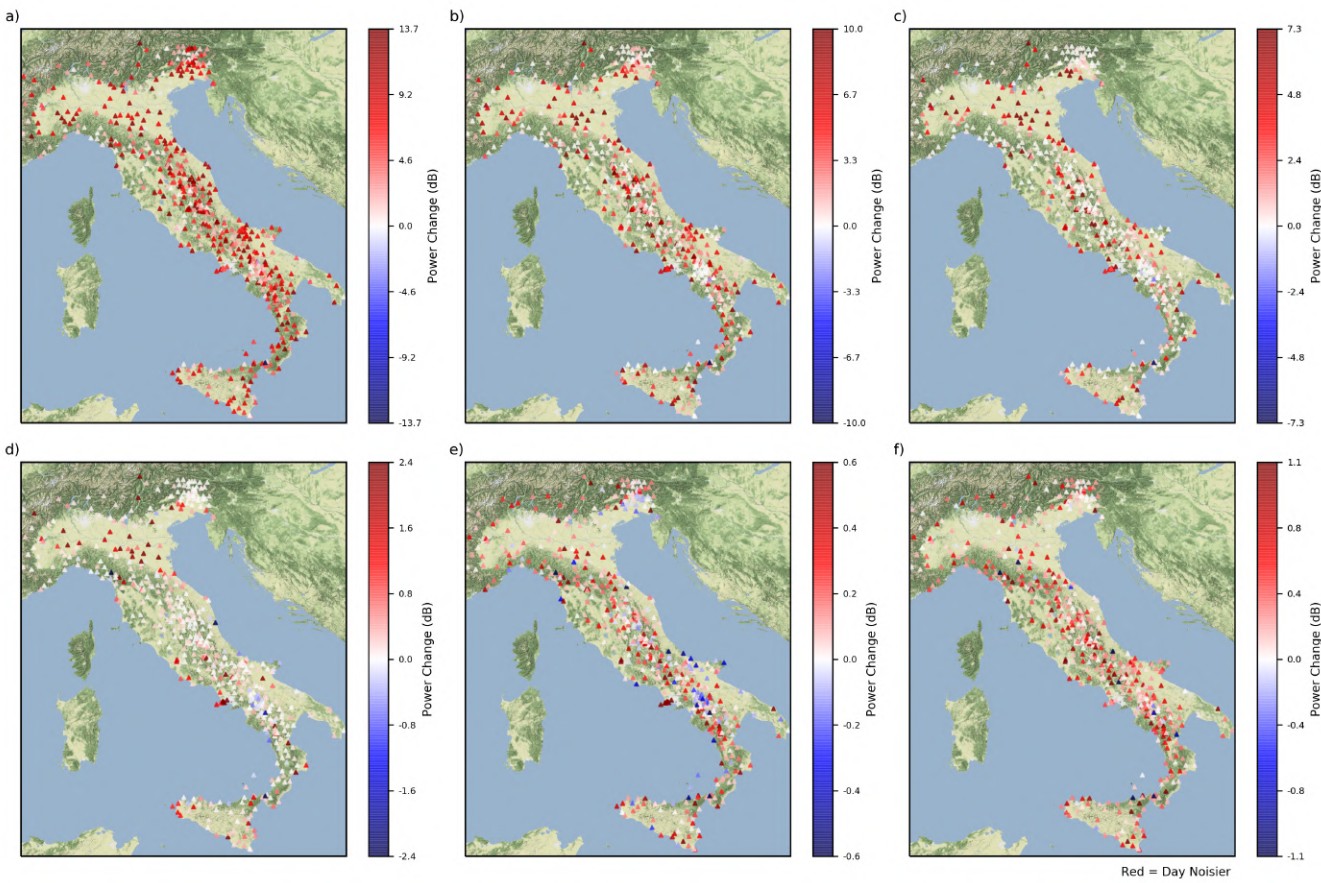

**Figure 6.** Median noise levels in dB for daytime and nighttime for the periods of $0.1\,\mathrm{s}$, $0.25\,\mathrm{s}$, $0.5\,\mathrm{s}$, $1.0\,\mathrm{s}$, $2.0\,\mathrm{s}$, and $5.0\,\mathrm{s}$. Red color means day is noisier than night. Basemap data are retrieved from © Stamen Design.

**Figure 7.** Median noise levels in dB for weekday and weekend time for the periods of $0.1\,\mathrm{s}$, $0.25\,\mathrm{s}$, $0.5\,\mathrm{s}$, $1.0\,\mathrm{s}$, $2.0\,\mathrm{s}$, and $5.0\,\mathrm{s}$. Red color means weekday is noisier than weekend. Basemap data are retrieved from © Stamen Design.

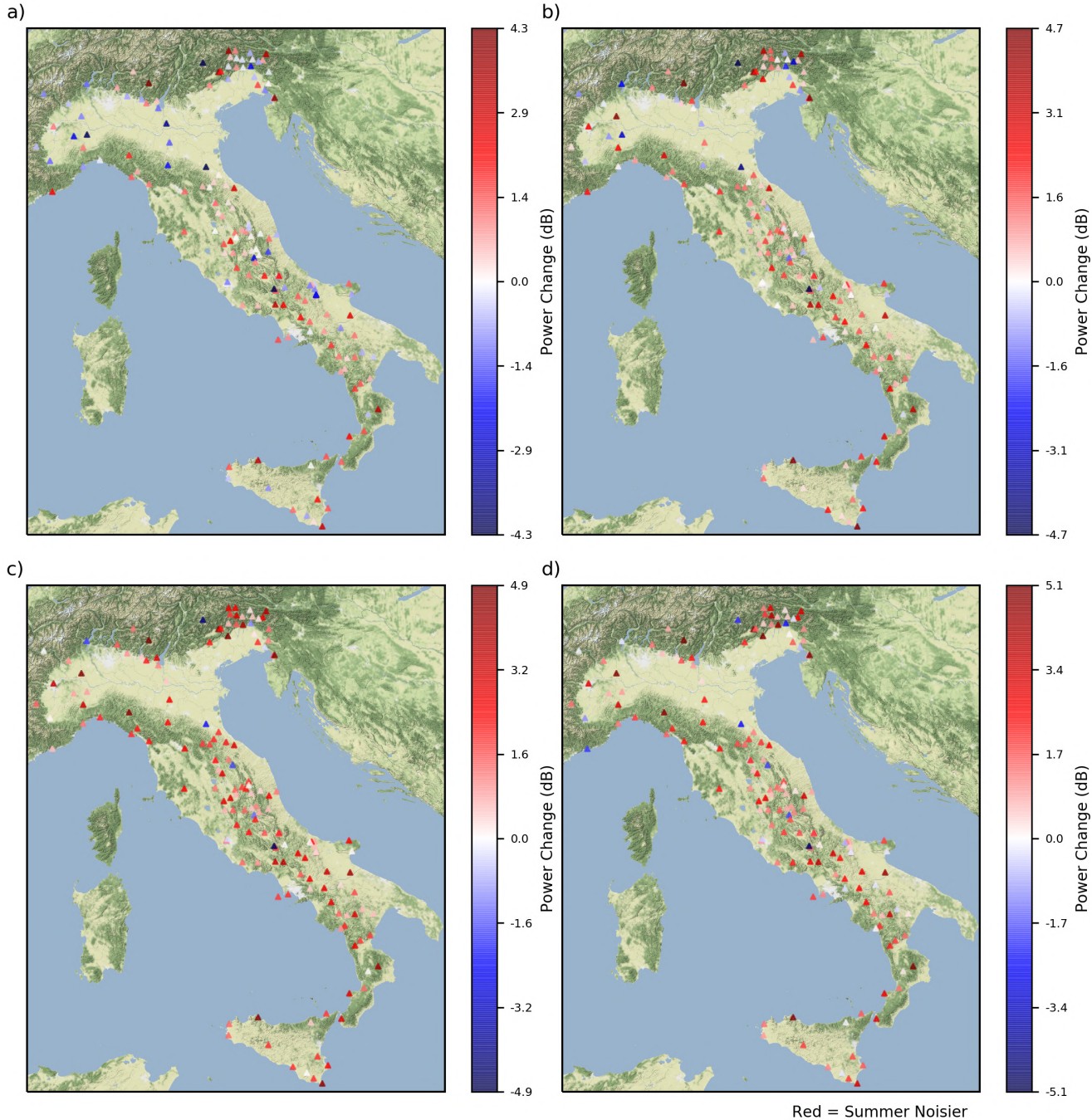

**Figure 8.** Difference between noise levels in 5 s, 8 s, 16 s, and 32 s between the winter and summer of 2019. Red color means summers are noisier than winters. Basemap data are retrieved from © Stamen Design.

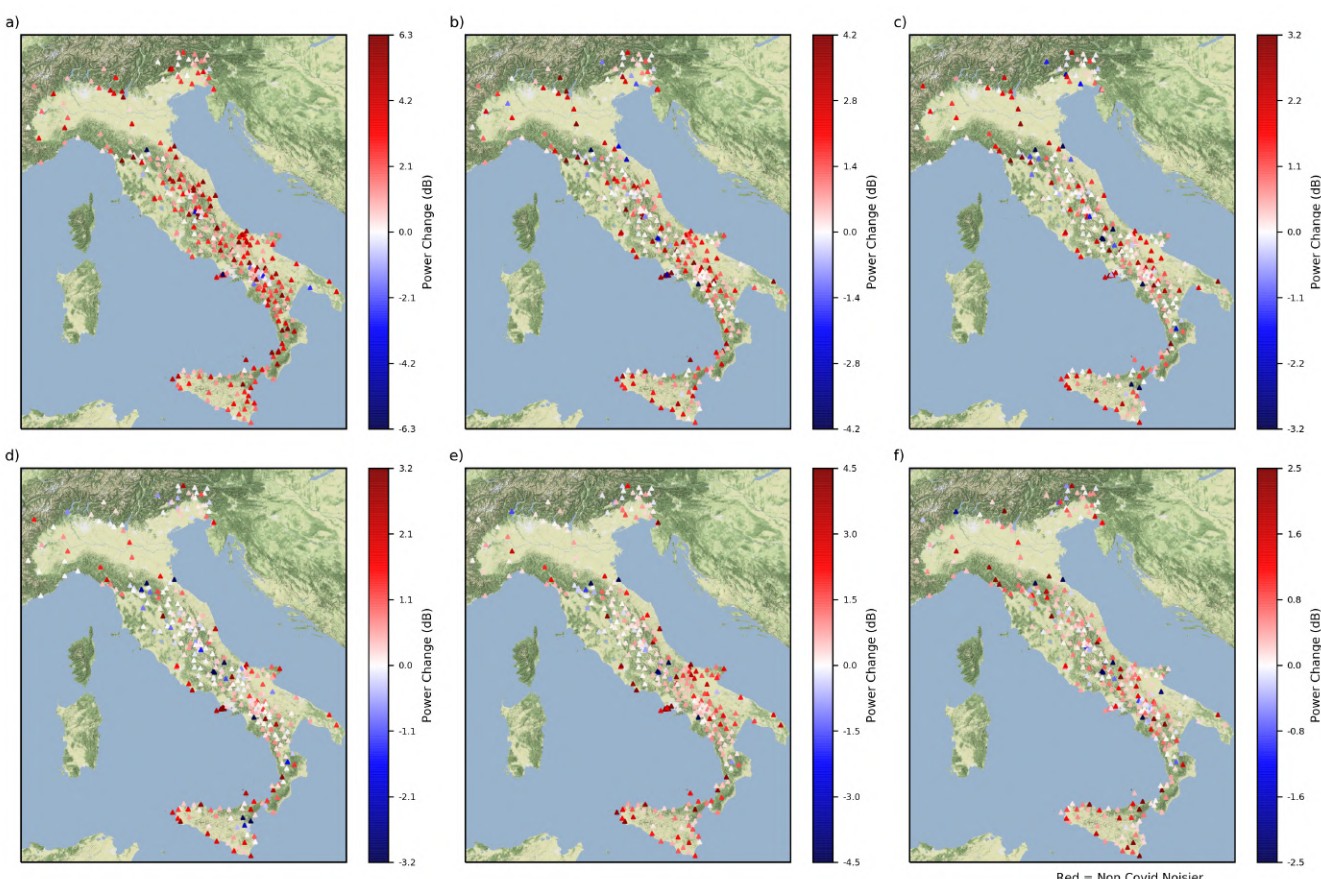

**Figure 9.** Difference between noise levels in 0.1 s, 0.25 s, 0.5 s, 1.0 s, 2.0 s, and 5.0 s between 2019 - 2022 and lockdown. Red color means 2019 and 2022 are noisier than lockdown period. Basemap data are retrieved from © Stamen Design.

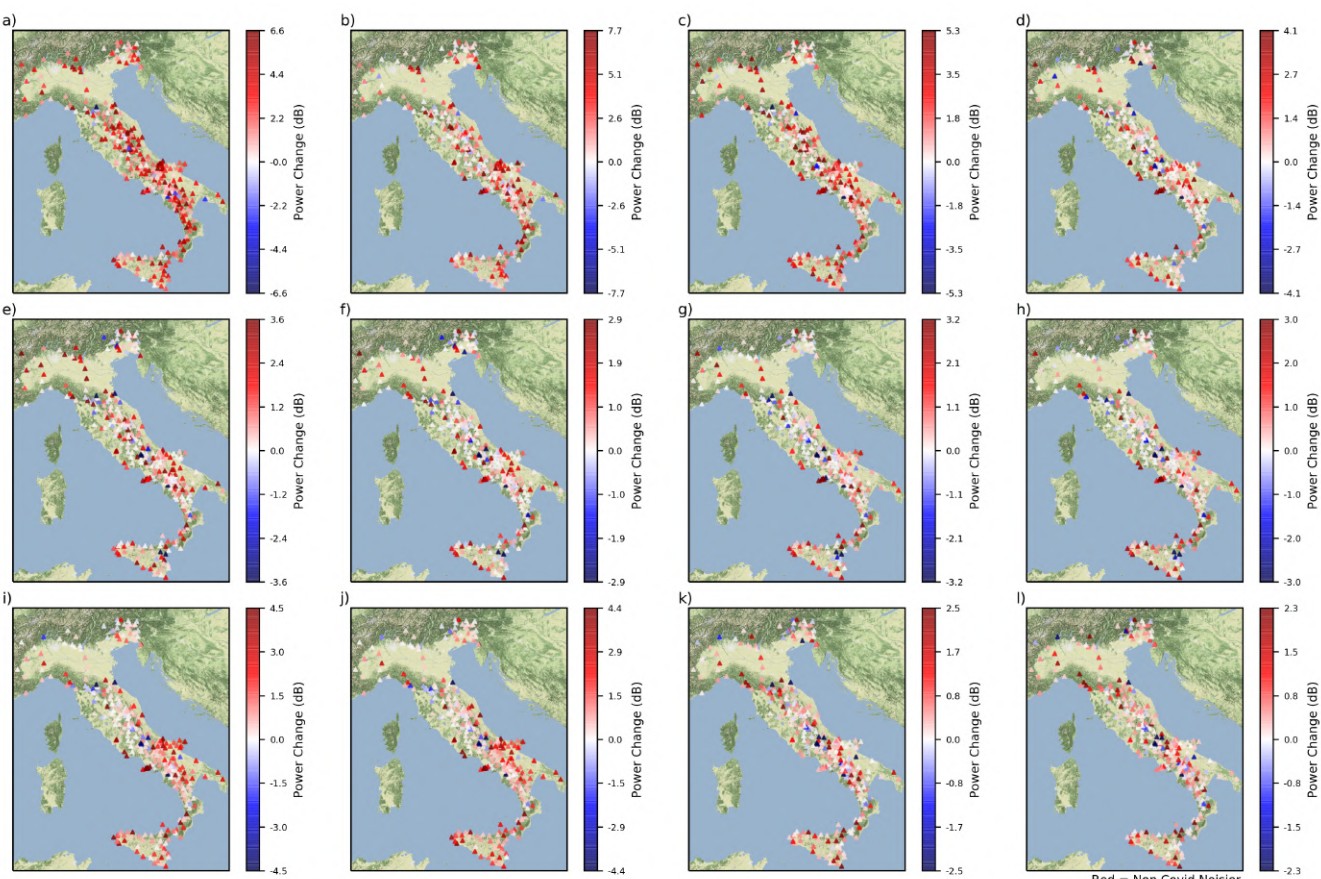

**Figure 10.** Difference between noise levels in 0.1 s, 0.25 s, 0.5 s, 1.0 s, 2.0 s, and 5.0 s between 2019 - 2022 and lockdown in a, c, e ,g, i, k) daytime, b, d, f, h, j,l) nighttime. Basemap data are retrieved from © Stamen Design.




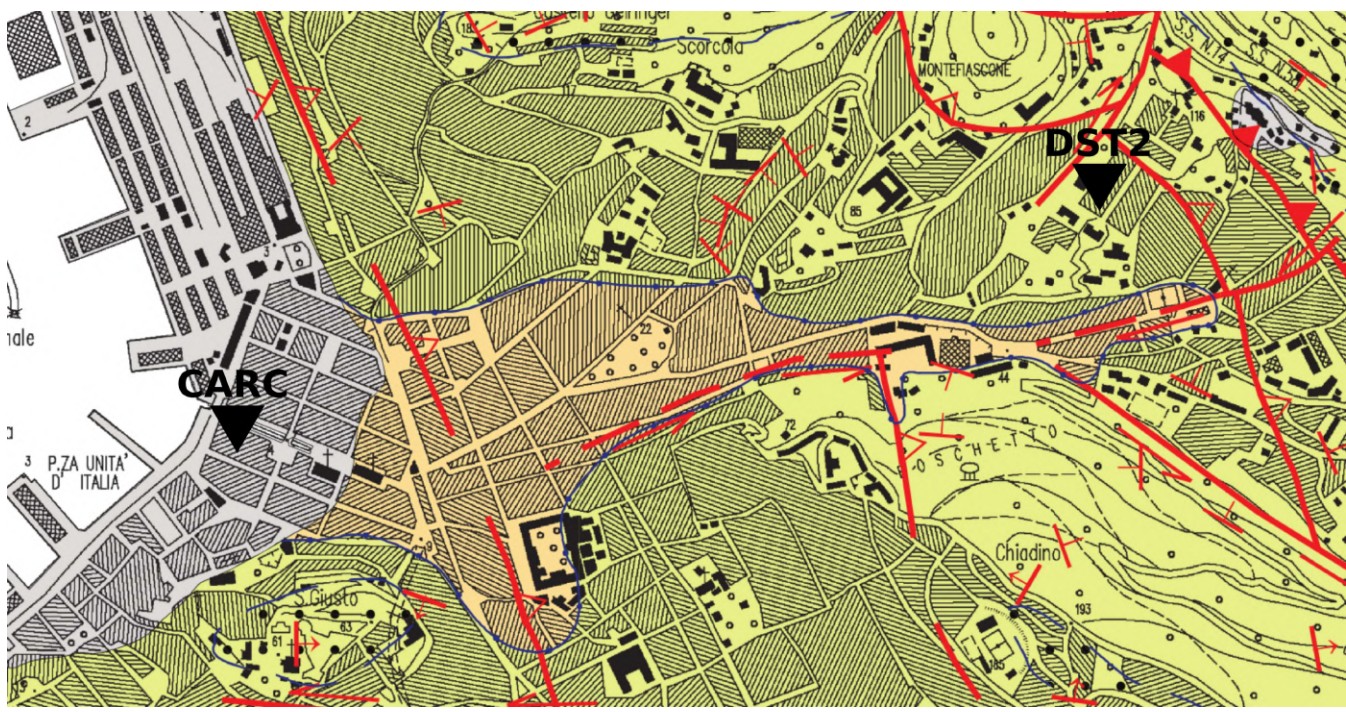

**Figure 11.** Geological Map of Trieste (grey, orange, and yellow colors indicate anthropic, ubiquitous deposit units, and flysh of Trieste, respectively), modified from Cucchi et al. (2013).

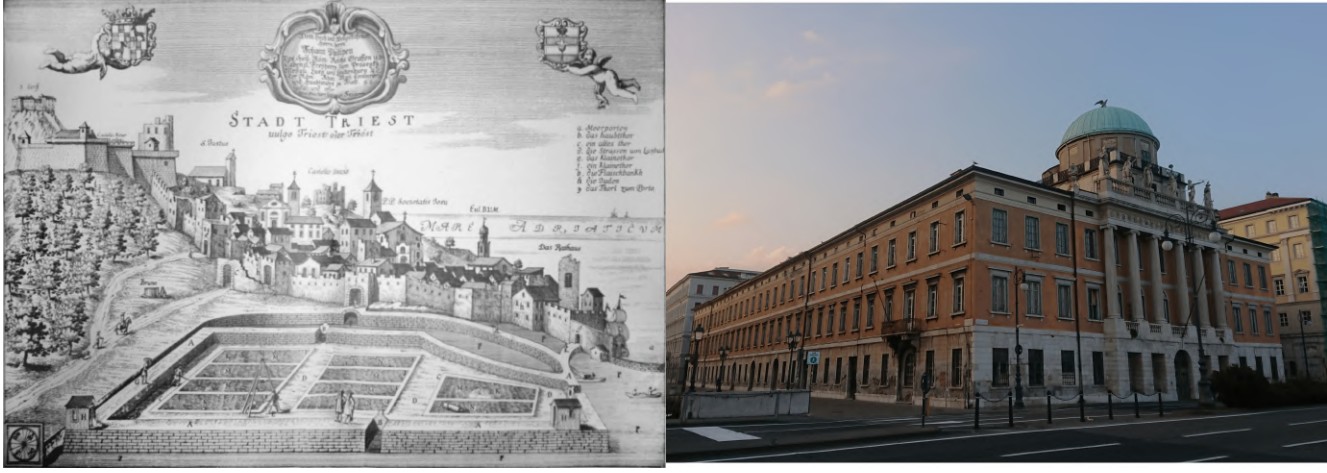

**Figure 12.** left) Drawing of the salina part of the city of Trieste by Johann Weikhard von Valvasor in 1689 taken from © Wikipedia (https://upload.wikimedia.org/wikipedia/commons/6/67/Mesto_Trst-Valvasor-2.jpg, last access: 7 November 2022), right) Palazzo Carciotti.





**Figure 13.** Hourly average plots of noise levels of DST2 (line) and CARC (line with dots) for no-lockdown (top) and lockdown (bottom) dates. ALNM and AHNM introduced by Cauzzi and Clinton (2013) are black line and dashed line, respectively.

**Figure 14.** top) Satellite image of the Palata. PLTA station is demonstrated with red triangle (latitude: 41.88, longitude: 14.78) station which is located in Palata municipality building in Central Italy (the image is generated by © Google Earth). bottom) Seismic record registered on 9th of May 2019. Three detected cars are presented in the upper right with red vertical lines presenting the initiation and termination position of the car signals.







**Figure 15.** top) Hourly average plot of noise levels of PLTA, and bottom) Average of FFT of the detected cars in PLTA in 2019.





| Land Usage | Code | Stations |
|---|---|---|
| Settlements | SL | 528 |
| Annual Cropland | ACL | 61 |
| Permanent Cropland | PCL | 20 |
| Grassland | GL | 46 |
| Forest | FL | 52 |
| Other land | OL | 8 |
| Wetland | WL | 0 |
| Water | WT | 0 |

**Table 1.** Land usage at the stations (Istituto Superiore per la Protezione e la Ricerca Ambientale, 2022).

| Parameter | McNamara and Buland (2004) D'Alessandro et al. (2021) | Anthony et al. (2021) | Present work |
|---|---|---|---|
| Window | 60min | 60min | 90min |
| Window overlap | 50% | 50% | 50% |
| Completeness | - | >90% | >90% |
| Sub-window | 900s | 819.2s | 900s |
| Sub-window overlap | 75% | 75% | 75% |
| Detrend | Linear | Linear | Linear |
| Gaps | Removed | Zero-pad | Linear interpolation |
| Window type | 10% cosine | Hann | Hann |
| Binning/smoothing | Yes | None | None |
| Average | Overlapped 1 octave | 1/3 octave | 1/3 octave |

**Table 2.** Data processing parameters for the evaluation of the PSDs of our study along with the studies of McNamara and Buland (2004), D'Alessandro et al. (2021), and Anthony et al. (2021).





| Period (s) | AHNM Threshold | No. of station | Percentage of network (%) |
|---|---|---|---|
| 0.10 | -91.50 | 58 | 10.90 |
| 0.25 | -101.34 | 43 | 8.08 |
| 0.50 | -114.06 | 88 | 16.54 |
| 1.00 | -118.53 | 183 | 34.40 |
| 2.00 | -111.20 | 41 | 7.71 |
| 5.04 | -97.66 | 8 | 1.50 |
| 8.00 | -104.91 | 12 | 2.26 |
| 16.00 | -104.14 | 40 | 7.52 |
| 32.00 | -102.60 | 51 | 9.59 |
| 64.00 | -99.53 | 67 | 12.59 |
| 80.60 | -97.93 | 64 | 12.03 |
| Any | - | 273 | 51.32 |

**Table 3.** Stations with higher than AHNM in the network.

| | Period (s) | | | | | |
|---|---|---|---|---|---|---|
| | 0.1 | 0.25 | 0.5 | 1 | 2 | 5 |
| db | 1.937 | 0.515 | 0.210 | 0.155 | 0.490 | 0.300 |

**Table 4.** Median noise level changes between $0.1\,\mathrm{s}$ and $5\,\mathrm{s}$ seconds. Positive values mean noise levels are higher during no - lockdown time span.