# Peer review of "Seismic Background Noise Levels in Italian Strong Motion Network"

_Natural Hazards and Earth System Sciences, 2022_

## Referee Comment (RC3)

**Seismic Background Noise Levels in Italian Strong Motion Network**
**by Simone Francesco Fornasari, Deniz Ertuncay, and Giovanni Costa**

Seismic data quality affects the earthquake monitoring capabilities of a seismic network significantly. More than that near-source strong motion records have great advantages for real-time estimation of earthquake magnitude by providing unsaturated recordings of moderate to large earthquakes to get fast and robust earthquake location and magnitude estimates. So, the quality of data from individual stations has to be estimated, evaluated and investigated constantly and regularly. In this framework, I found the submitted work of Fornasari et al. very valuable and precious. The results of the paper will be quite beneficial for the researchers who will use the data of the Italian Strong Motion Network. It is a necessity for each network to have similar studies investigating noise levels. So, I appreciate and encourage the submitted work and effort. I found this study, to have such an effort for Italian strong motion data, quite valuable. I would be in favour of the publication of this paper.

**Line 1:** "… with more than 700 stations." ITACA (https://itaca.mi.ingv.it/ItacaNet_32/#/station/search ) refers to 836 stations for Italian Strong Motion Network, FDSN presents (https://www.fdsn.org/networks/detail/IT/) 523 stations. May you please provide a specific number, with a source?

**Line 9:** "… we focused on relatively short periods (≤5 s), interested  **in** anthropic noises."

**Line 9:** You refer "**anthropic**" noise/activities/sources in Line 9, Line 24, Line 98, Line 119, Line 186, Figure 11 caption.

You refer "**anthropogenic**" noises/sources in Lİne 5, Line 42, Line 75, Line 107, Lİne 165, Line 223

You refer "**cultural**" noise in Line 15, Line 140, Line 159.

If you are using these terms in the same meaning, please be more consistent in the usage. If you are referring to different meanings, please describe this.

**Line 12:** "… stations are located in densely populated areas such as **the** center of Naples, …".

**Line 14:** "noise levels dropped to 6.5 decibels in **the** daytime and 12.5 decibels on weekdays. …

**Line 20:** "On the other hand, noise can be used for the characterization of layers of the earth (Shapiro et al., 2005), **moon (Larose et al., 2005) or Mars (Schimmel at al., 2021)**. "

Please refer also to those additional sources:
https://agupubs.onlinelibrary.wiley.com/doi/10.1029/2005GL023518
https://agupubs.onlinelibrary.wiley.com/doi/epdf/10.1029/2021EA001755

**Line 21-26:** Please cite also (**Stutzmann *et al.*, 2000**).

(https://pubs.geoscienceworld.org/ssa/bssa/article/90/3/690/102808/GEOSCOPE-Station-Noise-Levels )

**Line 48:** " that  **have** been carried out for more than 25 years."

**Line 64-Line67:** Please cite DOI's, for these networks properly:

Italian Strong Motion Network  - 10.7914/SN/IT -
Friuli Venezia Giulia Accelerometric Network - 10.7914/SN/RF
Irpinia Seismic Network - ?

**Line 75:** Regarding your commentabout the change of the stations on Line 51-Line52; I may suggest you to perform 2022-year noise as a whole, in order to present a full understanding of the background noise for the mentioned networks. Then, to compare the results with the Covid-19 lockdown period provide more meaningful and stable results.

**Line 112:** "Numerous stations exceed the levels defined by
Cauzzi and Clinton (2013)." May you please comment/discuss on the possible reasons of this situation.

**Line 126:** Diurnal and seasonal variations of seismic noise is well documented many years before **2021**. You should also check (**Stutzmann *et al.*, 2000**; McNamara and Buland, 2004), and please cite (**Stutzmann *et al.*, 2000)**, accordingly.

(https://pubs.geoscienceworld.org/ssa/bssa/article/90/3/690/102808/GEOSCOPE-Station-Noise-Levels )

**Figure 1:** Please mention what is "EM noise" and insert Cauzzi et al. (2014) to the References.

**Figure 4:** Does this caption refers to Supplement S1 or Table 4? Please, edit given period band in the caption accordingly. " Median vertical component noise maps in one-third octave bands around a-g) 0.1 s, 0.25 s, 0.5 s, 1 s, 2 s, 5 s, 16 s, 32 s, and 80.6 s."

---

## Author Comment (AC1)

The article under review cover an interesting topic and present some unprecedentedly published results, for these reasons I would be in favour for publication. Unfortunately, the manuscript does not fit the standard for publication and It requires, in my opinion: a deep review for different reasons:

1. the text is in some part confuse, with a lot of repetitions and for the reader (at least for myself) it is difficult to distinguish between original results, hypothesis from the authors and previously published results. It emerges clearly that this is the manuscript from a newbie researcher and I would encourage him to rephrase many sentences and drain the text as much as possible, to ease the readability and comprehension.

   Considering also the following points raised by the reviewer, we updated several instance in the manuscript trying to address the readability of the text.

2. a clear description of the methodology is missing. I understand that PSD is a standard but how PSDs were computed should be described, otherwise the results would be difficult to be reproducible. The authors do not mention is disturbances to the noise (e.g. earthquakes) are removed to the dataset.

   We updated the Method section with the information that were missing as pointed out by the reviewer.

3. Section results does not analyse in depth the results, e.g. figure 3 that contains the substance of the paper (PSD for single station, difference between the different months and years but it is not discussed at all.

   In Figure 3 only 20 stations out of 532 are presented to provide an overview of the behaviour of the noise over time. In the updated Figure 4 the overall noise levels of all stations are presented: the figure in the first version was a comparison figure for 6 periods instead of median vertical component noise maps of 9 different periods.

4. In Section Discussion I would suggest that the authors at first present their results and then they discuss them in the context of previous study. figures are difficult to be read. Italy is long and narrow and the authors are evaluating +500 stations that means +500 colored symbols placed in the map.

   In the Discussion section we divide the results in 3 categories (low, medium, and long periods). For low and medium periods we discuss day-night and weekday-weekend variations; for the long periods we discuss seasonal variations. The structure is inspired by Anthony et al. (2022). The results are then compare with the ones from D'Alessandro et al. (2021). Since COVID-lockdown is an extraordinary event, we discuss its effect in a separate section. The Trieste case study and the vehicle noise are discussed separately in their dedicated sections as they analyze very short periods (i.e. high frequency) that would be difficult to merge inside the more general discussion the main Discussion section. However, if the reviewer thinks that the Discussion section needs to be reorganized we can reshape it as below:

   5. Discussion

     5.1 Low periods

       5.1.1 Case study: stations located in Trieste

       5.1.2 Vehicle noise

     5.2 Medium range periods

     5.3 Long range periods

     5.4 COVID-19 lockdown

   For better visualization we provide HTMLs with interactive features in our GitHub repository (https://github.com/sffornasari/RAN-noise).

We are at the end of 2022 and the authors during the review phase will have the full 2022 year available. I encourage them to use that dataset to provide a much comprehensive analyse for a complete year for which a lot of stations should be available. and eventually to consider the option of dropping data from 2019 that could become less relevant.

We agree that adding 2022 would increase the quality of the paper. Right now, we are considering to do this. In the meantime we would like to wait for the response of the second reviewer before making such a major change in the paper.

Moreover,

1. L.20 To complete the though I would suggest to add that in this case earthquakes are considered as disturbances in the signal. ✓

2. L.21 Since the authors made a distinction, we now need a definition of what, for the case of this paper, is noise. ✓

3. L.25 I would suggest to add also the scattering at shallow layers that e.g. generates the so-called Newtonian Noise (e.g. Harms et al. 2009.) ✓

4. L42 "away from anthropogenic noises", I would say "far from any source of noise", usually seismometers are buried to prevent thermal fluctuation, and so on. ✓

5. L.44 Since the argument is faced in a general perspective, I would say that seismic stations are placed where it is appropriate for the purpose of the project itself. VBB stations are in remote places far from anything, accelerometer for site effects and strong motion are placed at the study site and so on.
We rephrase the paragraph highlighting the importance of the purpose of the network in the site selection criteria.
"Despite to optimize the quality of the recordings seismic stations should be installed away from any source of noise (e.g., roads, major cities, and factories), the selection of the "optimal" location to install a seismic station weights multiple parameters depending on the purpose of the specific network. The National Accelerometric Network (RAN) is established to monitor strong motions at a national level which is owned and managed by the Italian Civil Protection Department (DPC) gorini10,zambonelli11,costa2022near. The integrated RAN network is the combination with the following networks; i) the Friuli Venezia Giulia Accelerometric Network (RAF, Rete Accelerometrica Friuli Venezia Giulia in Italian, costa10) in the North-East Italy, owned and managed by the University of Trieste (UniTS) ii) Irpinia Seismic Network (ISNet, weber07) in the South of Italy, owned and managed by Analysis and Monitoring of Environmental Risk society (AMRA). From now on the RAN networks refers the integrated RAN."

6. L.50 I would suggest to extend this sentence. It would be difficult to understand why pandemic reduced the noise. Eventually including the citation of some of the paper on this topic as Lecoqe et al, Piccinini et al, Poli et al. ✓

7. L.54 "section 3 section" is a mistake ✓

8. L.57 Please note that COVID and COVID-19 are the same thing, same for "COVID lockdown" and "COVID-19 lockdown". please fix it using one name over the whole manuscript ✓

9. L.62 at line 46 RAN was called in a different manner "Integrated italian Accelerometric Network" In my view things should be called consistently along the manuscript
We rephrased the sentences introducing the integrated RAN (and its "sub-networks") deleting the definition given in Line 62.

10. L.63 At line 47 the contributors are differently described. If there is the need to repeat it, please be consistent.
We moved the description of the integrated RAN to the Introduction section.

11. L.65 "in the South" and "in the North East" please specify of what, South of Italy I suppose. ✓

12. L.66 I am getting confused, The authors use RAN as the acronym for the Integrated . . . ., then they write that the RAN is made by three networks. And one of the three is the RAN.
The integrated RAN network consists 3 different networks. We define what integrated RAN as shown above.

13. L.68 Again there is some redundancy in the description, the fact that some of them have been converted to continuous was already mentioned about. ✓

14. L.71 Third time the migration to continuous was mentioned.
We deleted the sentence.

15. L.73 at line 59, it is written that, for simplicity The authors will call it just lockdown. ✓

16. L.75 Piccinini et al, proved that this was not true at national scale.
In Figure 4 of Piccinini et al. paper there are several stations with almost no noise reduction. Stations such as RAVA (https://goo.gl/maps/DdCY2eZuePPQkkMAA) is located away from population centers as they also report in their paper. It is expected to have small to none noise difference during the lockdown since there are no anthropogenic noise sources nearby. We also reported several instances in our paper. However, we agree that the generalization that we made may not correct. Hence, we changed the end of the sentence to "[. . . ] were reduced in many places".

17. L.79 Question: data from 2021 would not be useful to integrate the dataset?
During 2021 different regional lockdown measures were activated by Italian authorities at different times. It would be hard to interpret the results. We decided to include on the data from 2019 and beginning of 2022 for this reason.

18. L.87 I would suggest to add a sentence describing the workflow to go from data (continuous time series) to PSD. e.g. data have been corrected for the response? How the spectrum was computes is not mentioned.
We added the details of the PSD calculation and the response removal process.

19. L90 I am not english mother tongue, but I feel that it is more appropriate to write "data" in place of "the data". Please check. ✓

20. L.104 better to say "described" if the author extend it, as suggested above, to a full description. ✓

21. L.104-105 figure 3 is not discussed although it contains THE RESULTS of the analysis. The reader cannot understand where the considered few stations are located and why they differ in noise level.
Figure 3 shows a portion of the stations that we have analyzed in the paper. Hence, it provides an overview of how the noise levels change over the entire dates that we considered. In Figure 4 (the updated one that we present below), on the other hand, you can see the overall noise levels for all stations.

22. L.105 This sentence is not clear, results are shown in fig.3, what is then in fig.4?
We rephrased the sentence explicitly addressing Figure 4.
"The results obtained for few randomly selected stations applying the method described in Section 3 are shown in Figure 3 for the periods of interest, namely 0.1 s, 0.25 s, 0.5 s, 1 s, 2 s, and 5 s: this provide an overview of the behaviour of the noise at different timescales for different periods, as described in details afterwards (see Figure 1). The overall background noise levels for the all stations in RAN presented in Figure 4."

23. L.107 I am feeling pedant but is RAN stands for Rete Acceleromentrica Nazionale, then it is not necessary to follow it by network. ✓

24. L.107-116 t the authors move from periods to frequency and backward. I understand that this is a common practice but, in a paper it is more appropriate to stick on one choice, otherwise the reader gets confused.
We replaced "frequency" with "period" in Line 112 and Line 116.

25. L.120 RAN stations at touristic sites can experience the opposite, quiet in the weekdays and noise in the weekend. Anthropic noise is very local. Did the authors consider it?
We checked the 'anomalous' stations with noisier weekends but we did not see a pattern with the touristic activity around those stations. Some of them are in small towns and villages. We assume that those areas are not hosting major touristic activities.

26. L.122 this is a repetition of line 119
We changed the sentence: "We also studied the changes in the noise levels between weekdays and weekends and the general trend of noisier weekdays are observed Figure 7."

27. L.123 english unclear to me
We replaced "are used" with a comma.

28. L.125 "very long period" please give the period band since for some seismologists this would be tens and even hundreds of seconds
We explained what we mean by "very long period" (i.e., >5s).

29. L.133 Since the author proved that a seasonal variation and a weeday/weekend variation exist, I wonder if they considered it when comparing lockdown and no-lockdown. I mean that, to be consistent and to catch only the lockdown effect, the comparison should be done only with the same time span of 2019 and 2022.
We believe that in the the lockdown period did not last enough to see any seasonal variations. We are in agreement with the reviewer about analyzing same time span to understand the exact effects of the COVID lockdown. However, for the sake of keeping the paper more compact, we did not include these analysis in it. But we decide add them to the supplementary material and it can be seen in Figure S2. We also refer these figures in the paper.

30. L.140-143 Sentence is too vague
We changed the sentences:
"Table 1 shows the distribution of the stations according to the classification proposed by ISPRA. Despite most of the stations are located in urban areas, and then been potentially subjected to high levels of anthropic noise, this classification is too reductive (e.g., not considering the population density and the

presence or making a distinction between residential and industrial areas) to be associated to specific noise levels."

31. L.144 and following, Since the effect is local, did the authors consider the eventual presence of Wind Farms, or other facility that could produce anthropogenic noise at longer periods?
We did not analyze the effect of the wind farms since we do not have the list of wind farms in Italian territory.

32. L.149 "is assumed" It is not an assumption, it is an observation from data and from road traffic data, mobility from mobile phone records and so on. There is plenty of data showing that human activity is reduced. ✓

33. L.153 "trend" I would say pattern. ✓

34. L.153 "of" typo? ✓

35. L.155 "if a station is located in a settlement" I would expect that this is one of the result of this study, not and hypothesis within the discussion section. is this observed in data or not?
In the high frequency the sources of the noise are linked to the human related activity. Hence, we present a very straight-forward explanation to it. However, we do not analyze each factor that may play a role on the high frequency noise source. Because of that, we presented only a possible explanation for the low weekday- weekend difference in noise levels. Nevertheless we changed the sentence and deleted the parts where we address the sources of the noise specifically, referring them to human activities.

36. L.158-160 Again, do the authors observe what described by other authors in their analysis? This is not a review paper but a scientific one.
The cited studies provide a justification for the type of analyses that we performed and also identify the noise sources in the specific period range, which is beyond the scope of our paper. In the following sentences, we provide a description of the behaviour of the noise both daily and weekly.

37. L.169 "stations start". Start means that there is a variation to me, when do they start? No clear.
We changed the sentences: "Considering weekly variations, stations become noisier on weekends with decreasing power change."

38. L.175 If the last sentence applies, that implies that stations are blind in this range of period. I do not understand why discussing the source of noise when in this frequency band accelerometers just measure the self noise of the instrument. Moreover the self noise can be computed and measured. It is not a matter of believing. Am I wrong?
We agree the reviewer about the usage of the word "believe" and rephrased the sentence: "As indicated by Cauzzi et al. (2013) the main source of long period noises in the case of accelerometric recordings can be associated to the instrumental noise of the RAN stations."

39. L.179 Again the authors discuss something that they cannot observe. I suspect this depends on the fact that they are using accelerometers and D'alessandro et al. (2021) used velocimeters.
The two main factors that contribute to the different results with respect to D'Alessandro et al. are the different types of instruments used (we use accelerometers and they used velocimeters, BB?) and the position of the stations used over the territory (in settlements vs far away from them). We can specify this. That being said we believe it's a relevant result, especially considering that no previous studies have been carried out on continuous recordings from accelerometers (in Italy at least).

40. L.185 "period periods" repetition ✓

41. LInes 185-189, in summary:

 (a) human activity dominate noise in this freq band.
 (b) high noise can be linked to activities
 (c) less human activities less noise.

Do we need a scientific study and a paper to state this? Different is when this is a direct observation from data. but this is not what the authors write in these 2 sentences.
This is a direct observation from our data. This is the first time the integrated RAN's noise levels are presented. We would like to analyse its noise content even if some of the results may be guessed without even looking the actual data. By giving the noise level changes in day night and weekday weekend, we also present the contribution of human and/or other sources in the background noise levels of RAN.

42. L.190-194: COVID reduced human activity, ok. Human activity influence seismic noise. Noise is higher in populated areas and near buildings. A dozen of paper noted such a reduction. The authors too. OK, what is the added value of this study for the COVID-19 lockdown ? It is not clear to me.
We agree that there are numerous studies published specifically for the COVID-19 lockdown. We point out though that no other paper used the RAN data. Thanks to the idea behind the RAN,capturing ground motion information in the urban areas, study the COVID-19 lockdown allowed us to evaluate the background noise level of the network with reduced human related noise.

43. L.215 "are" should be "is" ✓

44. L.216 "at" should be "in" ✓

45. L.222 "dates" ???? I presume median of the PSD noise.
We changed the "dates" with "periods".

46. L.223 "are more dominant" could be "prevail"? ✓

47. L.233 I miss to understand how this paragraph, at the end of section discussion is linked to the rest of the study. It would make sense at the beginning of the analysis when authors tackle the problem of distinguising between different source of noise and of characterize their frequency content.
We add a sentence to the Section 5.2 explaining the choice of the two particular situation: "The selection of these two particular stations is due to the extensive knowledge about their spatial and administrative information."

48. L.235-237 I suggest to rephrase the sentence. ✓

49. L.240 "manually" "by hand", As I mentioned, I am not english mother tongue. But to me, this sentence means that somebody was checking the passage of cars using his own hands. Not that, as I suppose, somebody visually inspected seismic data and searched for the effect of the passage of the cars and manually marked it on the seismic trace.
We changed the sentence to "visually analyzing the data".

50. L.247 "is" should be "are" ✓

51. L.254 the assertion "capable of providing . . . ." Is a qualitative speculation not based on true values. can the authors give some estimate of the miminum magnitude that can be detected at local distance by high noise and normal noise accelerometric stations?
We choose the 10 most noisiest stations to understand their capabilities on P-wave corner frequencies defined by Brune (1970).

52. L.256 ". . . but also the small ones" This sounds a bit obvious and not very useful without, as above, an estimate of the detection capability. Big and small are always relative to something.
We agree with the reviewer that some of the sentences in that paragraph is unnecessary. Thus we modified it with the comparison with Brune's model (1970).

53. L.258 "they" I cannot understand who is the subject: Selection criterion for what?
Here "they" is referred to median noise levels. The selection criterion depends on the nature of the study. For instance, one can exclude stations with high noise levels, if the research will be about small magnitude earthquakes. On contrary, noisy stations can be useful for benchmarking seismic denoising methods.

54. L.259 "Some of the stations" How many? again description of data and of result is too vague for a scientific paper
In total 81 of our stations are installed inside a building. We added the information to the paper.

55. L.260 "(528. . . . ) whereas some of them" In the data description it is written that the study is based on 528 stations. If 528 are in settlement, how can be that some of them are away from settlement?
We modified the sentence by deleting this information.

56. L.263 "in the short period" could be "in the short period band"? ✓

57. L.273 and following. This is a repetiition of line 172 and following
We modified the sentence.

58. L.281 ".. is applied" not clear
We modified the sentence: "During the COVID-19 lockdown in Italy (from March to May 2020), [. . . ]".

59. L.296 Anthony et al 2021 was published in 2022. ✓

60. Figure 1, I was surprised to see that Anthony et al. (2021, actually 2022) report info for only such narrow band and I checked the paper where for example I read (last paragraph, second column, pag 648) that noise in the band 0.0625-1 second contains cultural noise. So the narrow band should be as large as covering the entire band. Please check also the other.
In Anthony et al 2022, the interpretation of their results are done for very narrow frequency bands (shown in Figure 1 of their paper). However, as reviewer mentioned, they also provide frequency ranges from various noise types. We updated our figure accordingly.

61. Figure 2a, by placing the closeup box over Sardinia, the reader misses to appreciate the network coverage in that portion of the study area that is the whole Italian country.
The placement of the closeup box over Sardinia doesn't affect the visualization of the network coverage as there are no station there.

62. Figure 2, I wonder if there is a reason to plot stations with reverse triangles while in Figure 4 are not.
In Figure 2 all networks have their dedicated markers whereas rest of the figures have triangles for all stations. As reviewer said triangle marker for IT network may create misunderstanding. Hence, we changed the marker type for IT to diamond.

63. Figure 2 caption, the color coding of the figures is not descriptor. Moreover I do not understand what (RF) stands for.
The colorbar provides the information about the data completeness that are shown in the 3 subfigures. Subfigures are for 2019, 2020, and 2022. In subfigure a) and b) we zoom in several parts of Italy to show ISNet (IX) and RAF (RF) networks which are part of the integrated RAN. Definition of RAF has been provided in line 48. However, we acknowledge the fact that, the figure is complex and we did not provide all necessary information in the original caption. Therefore, we update the caption of the figure to : "Data availability of the stations in a) 2019, b) lockdown period, and c) 2022. In a) the close up box highlights ISNet (IX) and in b) the close up box highlights RAF (RF). Basemap data are retrieved from © Stamen Design."

64. Figure 2 palette:I I read PSD database ompleteness does this means that the authors counted the expected number of PSD for e complete time-series and then computed the ration of available ones?
As said by the reviewer, the completeness is referred to the number of computed PSDs with respect to the theoretical total number for the specific time range.

65. Figure 3, caption says "several stations" while in the manuscript I read few stations and actually there are a very small fraction of 528.
We specify the number of stations in Figure 3 (which is 20).

66. Figure 4, in the caption the authors use "Power Change" while in the caption and at line 113 I read "noise". Since "power change" is introduced in the discussion and not in the caption I do not understand what figure 4 display.
In Figure 4, the wrong figure wass presented by mistake. We changed the figure and it can be seen in the updated version of the paper. We also put the same figure below

[Figure]

Figure 1: Median vertical component noise maps in one-third octave bands around a-g) 0.1 s, 0.25 s, 0.5 s, 1 s, 2 s, 5 s, 16 s, 32 s, and 80.6 s. Upper and lower limits of the color bar are defined by the model developed by cauzzi2013high. Vertical components are presented in the following figures and Electronic Supplement. Background noise levels of all calculated periods can be found in Figure S1.

67. Figure 8, seasonal variability. In the manuscript it is mentioned that: 1) seasonal variability is studied only for year 2019 and it makes sense since data coverage for 2022 is limited to January-April. It is also mentioned that data analysis is limited to stations with completeness above 90% and it also make sense. In figure 8 I see only two triangle in Pianura Padana and one of them in figure 2a is colored in green that means  40% to my understanding. Apparently something is not correct. I wonder is this applies also to other stations.
90% completeness is referred to the 90 minutes window used to compute the PSDs. However, as the reivewer mentioned, it is important to have sufficient amount of data to calculate seasonal variations. Hence, we introduce 50% completeness in the days that we have records. We updated Figure 8 accordingly and explain the procedure as: "Stations with more than 50 % of data for both summer and winter time periods are selected to analyze seasonal effects."

68. Figure 12, the authors did not provide indication of where Trieste is.
Figure of the left is the drawing of part of Trieste. Photo on the right panel shows a building in Trieste. The building hosts the CARC station which can be seen in Figure 11.

69. Figure 13, If I interpet correctly this figure line for 00:45 $\pm$ 45 (purple with dots) has high noise at .1 seconds, while line for 23:15 $\pm$ 45 (red without dots ) has low noise. How can be midnight much noiser than 11pm? This contractics expectation described in the manuscript. Am I wrong?
CARC station (line with dots) has noise level around -95 dB (we add the label for y-axis) at 00:45 $\pm$ 45 whereas at 23:15 $\pm$ 45 it has noise level around -90 dB which means in 23:15 noise levels are higher than 00:45.

70. Figure 14, "demonstrated" better to say locatized
We changed the sentence.

71. Table 1, It confuses me. Since the total of stations gives 715 but the authors used only 528 of them. Is it revenant this table ?
RAN network has 715 stations but only 532 were in continuous mode (including the ones converted from triggered to continuous recording between 2019-2022) which are eligible for our analysis. As this may create confusion, we reduced the information in Table 1 to the continuous station, i.e. the stations used for the current study.

| Land Usage | Code | Stations |
|---|---|---|
| Settlements | SL | 388 |
| Annual Cropland | ACL | 48 |
| Permanent Cropland | PCL | 12 |
| Grassland | GL | 39 |
| Forest | FL | 38 |
| Other land | OL | 7 |
| Wetland | WL | 0 |
| Water | WT | 0 |

Table 1: Land usage at the RAN stations (ISPRA).

---

## Author Comment (AC2)

The manuscript by Fornasari et al. is original and intriguing because it faces different issues, among others the quality of the seismic networks and the origin of the seismic noise. Moreover, it shows the effect of the COVID-19 lockdown in Italy on noise, which was expected but never seen in consideration of the unicity of the lockdown period.

Nevertheless, the writing and the general organization of the work are critical. In my opinion this manuscript needs major revisions to be published. What I found to be critical in general is that the writing is really confusing, there are lots of repetitions, refuses, some useless or appended sentences and different jumps in the order of the communication. The result is that the reader cannot understand what the authors want to say. I would encourage the authors in publishing (especially if newbie researchers as at least the main author is) with a point-to-point revision of the text, but sorry I have no time to do that. However, I try to point out and suggest something as follows:

1. the stations used for the analysis are not very well presented. The authors use RAN for the real RAN but also for RAF and ISNet. I suggest to be very clear in the paper or to figure out something that does not actually exist, a sort of Integrated National Accelerometric Network (INAN). I don't know if this latter it's a good strategy: my concern is that we are not considering the INGV accelerometric stations, so we cannot define "National" the integrated network. By the way, why didn't the authors consider the INGV stations?

   The National Accelerometric Network (RAN) is established to monitor strong motions at a national level which is owned and managed by the Italian Civil Protection Department (DPC). The integrated RAN network is the combination with the following networks; i) the Friuli Venezia Giulia Accelerometric Network (RAF, Rete Accelerometrica Friuli Venezia Giulia in Italian) in the North-East Italy, owned and managed by the University of Trieste (UniTS) ii) Irpinia Seismic Network (ISNet) in the South of Italy, owned and managed by Analysis and Monitoring of Environmental Risk society (AMRA). The term "integrated RAN" has been historically used to define the combination of these networks. INGV stations are not included as a part of the study for 2 reasons, i) seismic background noise of INGV stations is studied quite recently by Antonino D'Alessandro (`https://doi.org/10.1029/2020EA001579`) and ii) our working group has full access to DPC's database. Hence, we would like to present only the data coming from DPC. However, we are glad to collaborate with other Italian seismic networks in the future to create a more complete background noise models for the Italian territory.

2. Also the number of used stations is not very clear. RAN consists of 647 digital stations, RAF of 14 stations and ISNet of 31 stations. Overall, there are 692 accelerometers, no "more of 700" as reported in line 63. The authors say that they used 528 stations because many of them still run in trigger acquisition mode and this number changed over time. So, what should be clear is: how many stations did the authors use in 2019, 2020 and 2022 for the analysis? Figure 2 is difficult to read (by the way, what do the different colors mean in the Figure?) and it is poorly commented.

   We update the information related with the stations so that it is easier to understand the information related with stations. The RAN consists of more than 700 stations of which 532 provided continuous data in the time range that we are interested in. Over the years we have 241, 325, and 526 stations for 2019, 2020, and 2022, respectively. We updated Figure 2 and it can be seen in below. In the updated version diamonds, stars, and circles represent RAN (IT), ISNet (IX), and RAF (RF) networks, respectively. Colorbar presents the completeness of the data. diamonds represent the RAN network regardless of the completeness values. In the zoomed frames, ISNet and RAF networks are given different colors (star for IX and circle for RF) to explain where the stations are located.

[Figure]

Figure 1: Data availability of the stations in a) 2019, b) lockdown period, and c) 2022. The close up boxes in lower left and upper right highlight ISNet (IX) and RAF (RF), respectively. Basemap data are retrieved from © Stamen Design.

3. Table 1 is also confusing about the stations actually used (715 stations are reported) and in my opinion is useless. A sentence in the text is enough to describe the deployments.
   Number of stations presented in Table 1 are updated limiting the information presented only to the continuous stations, as it can be seen below. We believe that providing this information as a text would be harder to follow. Hence we would like to keep it as it is.

| Land Usage | Code | Stations |
|---|---|---|
| Settlements | SL | 388 |
| Annual Cropland | ACL | 48 |
| Permanent Cropland | PCL | 12 |
| Grassland | GL | 39 |
| Forest | FL | 38 |
| Other land | OL | 7 |
| Wetland | WL | 0 |
| Water | WT | 0 |

Table 1: Land usage at the RAN stations.

4. the paper focuses on the accelerometric stations and the authors show several results for different frequencies (periods). The authors should mention that the accelerometers are not very sensible to low motion, such as noise, especially at low frequencies. Then it's normal that the low-frequency noise recorded by the stations could be out of meaning.
   We agree with the referee. This is why only in Figure 4 and Figure 8 longer periods are briefly mentioned. Periods between 0.1 to 5 seconds are the main interest of the study.

5. the method of computation is merely cited and the differences between the real used method and the standard ones are only reported in Table 2, but not too much discussed. In my opinion the authors should at least present the formula of PSD, the data preprocessing (e.g. instrumental correction, the kind of spectra) but also clearly explain their choices. In other words, the authors should answer these questions: what are the improvements using longer windows and the linear interpolation for gaps? Why don't they use the standard computation for PSD?
   We agree that the method, although standard in this field of study, needs a more in-depth description. We provide a more detailed description of the operations performed. The choice of the analysis window length is within the values commonly used for this kind of study (normally ranging between 1 h to 3 h): as Anthony et al. (2020) noticed, the length of the window became less relevant the shorter the periods of interest are. Our specific choice of using 90 min windows is motivated both by a trade-off between temporal resolution and memory requirements and by the fact that allowed us to perform the computation over the sub-windows without leaving data out (unlike in Anthony et al., 2022).

6. Another point is that there is no word on how the authors took into account the earthquakes or other strong transients that occurred in the time series. Have they been cut off or maintained?
Transient signal, consisting also of earthquakes, are not removed from the seismic traces since they are low-probability occurrences with respect to ambient seismic noise (McNamara and Buland, 2004, `https://doi.org/10.1785/012003001`): even though Anthony et al. (2020) showed that earthquakes can affect the noise level significantly for long periods (>10 s), they also concluded that this their effect on shorter periods (i.e., the main focus of our study) is negligible.

7. the Results Section is not very well explained and organized. Figure 3 and then Figure 4 are presented as "the representative noise level" of some (or all) stations. Then there is a (very short) discussion about the day-night, weekend and seasonal variations of noise, variations that can be important. So, I am a little bit confused about what the authors consider "representative": is it the night level, is the weekend level, or what?
Figure 3 shows the overall picture of several stations among years. Their medians are presented in Figure 4 (see below). In Figure 3, one can follow the difference between covid lockdown and non-covid time range. Moreover, weekday-weekend differences can be followed. However, it is unlikely to see the day night difference. Day-night, weeekday-weekend, and covid no-covid information are further analysed in multiple figures.
Day-night variations are presented in Figure 6 and discussed in Discussion section. Weekday-weekend differences are presented in Figure 7. We are only capable of making broad interpretations of the day-night and weekday-weekend variations. For instance, in Figure 6a, there is an overall trend of noisier day. But there are some stations which do not have large day-night differences like the others. Many factors may play role role such as population density, number of residential/industrial places around, car density of the roads nearby etc. Likewise, in the weekday-weekend difference, tourism activities may play role but we do not know the tourism density and their variations over time.
The word "representative" in line 107 may cause a false interpretation. By representative, we mean for each period that we calculated the value that we get is nothing but noise. As the reviewer mentioned, strong motion recorders may not cover the long period signals properly. This is why in almost all analyses we did, we do the interpretation up to 5 s.

8. Figure 4 is the most important one, and in my opinion deserves more discussion. Unfortunately the figure (and this is the same for all the figures like this) is very hard to be read. In this case, I suggest extracting from Table S1 the first and last 10 stations ordered by the noise level, it could help in the interpretation.
In Figure 4, the wrong figure was presented by mistake. We changed the figure and it can be seen in the updated version of the paper. We also put the same figure below. Unfortunately, it is really challenging to present all the stations and periods in a single figure and keep the readability high. To overcome this problem, we prepared an HTML version of the figure where zooming is possible. It can be found in the GitHub repository of this study `https://github.com/sffornasari/RAN-noise/tree/main/HTML`.

[Figure]

Figure 2: Median vertical component noise maps in one-third octave bands around a-g) 0.1 s, 0.25 s, 0.5 s, 1 s, 2 s, 5 s, 16 s, 32 s, and 80.6 s. Upper and lower limits of the color bar are defined by the model developed by Cauzzi and Clinton 2013. Vertical components are presented in the following figures and Electronic Supplement. Background noise levels of all calculated periods can be found in Figure S1.

9. Another point is Figure 5. What is the meaning of it? In my opinion it's a general overview of the entire "integrated" network but it's not very useful to know that the network is "good" in average.
   PSD probability density function can be considered as a standard procedure for the network operators to understand the quality control of the stations/networks. We would like to present the overall status of the integrated RAN network. This may help readers to compare the integrated RAN with other networks in a 'standard' way.

10. Figure 6 and 7: it's not very clear if the colour scale represents (as for Figure 8) the difference in noise levels between supposed calm (night, weekend) and disturbed (day, weekday, winter) periods.
    In Figure between 6-8 noisier day time, weekday, and winter are represented with red colors. Meaning of the colors are provided in the lower right of Figures 6-10.

11. Moreover, the authors present the PSD analysis for the COVID-19 lockdown period, but it's merely a list of Figures, without comments on them. In general I think that in this section the authors must clearly illustrate and comment on the results.

There is a dedicated section for COVID-19 lockdown period (see Section 5.1 COVID-19 Lockdown). Further analysis related with the temporal and spatial noise level changes related with the lockdown period such as the correlation between the noise level reduction and population and car density, tourism activity and so on but this is not the scope of the study. Hence, we provide a general overview about the effects of the lockdown on background noise. In Figure 9 and 10, background noise difference between lockdown and other dates are quite visible, as expected. To do more detailed analysis, we need many details that we do not have. Hence, we are only able to provide overall results.

12. The Discussion Section (also the others but this in particular) requires a deep English revision.
To overcome some repetitions, we made numerous changes in not only in Discussion but in all paper.

13. The number of analysed "noise levels" is reported as 525(!).
We assume the referee refers the line 146. In this line, it is written that "In 273 stations of 525 noise levels exceed the AHNM developed by Cauzzi and Clinton 2013 (Table 3).". In other words, 273 stations out of 525, i.e. the total number of analysed stations, have at least one period that exceeds the AHNM developed by Cauzzi and Clinton 2013.

14. In the discussion about the sea, swells and/or wind effects on noise, the authors should take into account that the accelerometers are not the best kind of sensor to record these low frequencies (high periods).
In the dedicated paragraph (line 170-175) we mentioned that the long period noises can be associated to the instrumental noise.

15. Lines 173-175 are a good example of what I mean with "jumps in the order of communication" and/or appended sentences. In the long period the accelerometers are completely deaf.
We rephrase the mentioned sentences and they can be seen in below, "As indicated by Cauzzi and Clinton (2013) the main source of long period noises in the case of accelerometric recordings can be associated to the instrumental noise of the RAN stations. As a proof of it, unlike in D'Alessandro et al. (2021) in which the smooth transaction from coastlines to the inlands and mountains are visible, in our study there is no change in noise levels from shores to inland and from high altitudes (Alps, Apenines) to low altitudes (Po valley)."

16. Table 4 refers to a particular station? Or is an average?
It refers to the median noise level changes of all stations for different periods

17. "Changes in the daytime are more significant than the changes in the nighttime between the lockdown and no-lockdown time span." How do you explain that if the working activity was almost zero during the lockdown?
We interpreted this results as a fact that, being the nighttime already quiet than daytime during ordinary periods, the effect of lockdown measures did not affect the nighttime noise levels as much as the daytime ones, when on the other hand the ordinary activities were severely limited.

18. I think that many comments should be moved to the Results Section.
We assume that these comments are from Discussion section. If we move some of our comments to discussion, this may create a paper harder to read due to separation of the interpretation of our results. If the reviewer can be more specific about the "comments", we can re-evaluate the structure of the text.

19. I don't really understand the utility of introducing here sub-section 5.2. Maybe, but I am not sure, it could be moved in the Result Section after or together with sub-section 4.1.
As we suggested to the first reviewer, we can re arrange the discussion sections as below,

    5. Discussion

        5.1 Low periods

            5.1.1 Case study: stations located in Trieste

            5.1.2 Vehicle noise

        5.2 Medium range periods

        5.3 Long range periods

        5.4 COVID-19 lockdown

20. Also sub-section 5.3 has nothing to do with the Discussion Section, maybe it can be inserted in the Result Section but the authors should introduce the problem.
As we write above, we consider re-organizing the sections of the paper.

21. What is missing is some consideration about the general quality of the sites of deployments of accelerometers. On the basis of the results of this paper, what are the effects on the strong-motion monitoring in Italy? I know that it's a very wide answer, but an effort to answer should be done.
    We added a new graph to supplementary material by plotting the 10 most noisiest stations to understand their capabilities on P-wave corner frequencies defined by Brune (1970). Similar critisim is done by the Anonymous Referee 1 and (https://doi.org/10.5194/nhess-2022-258-AC1). This may give an insight of the capabilities of the "worst" stations in terms of background noise. Overall status of the network is explained by Costa et al. (2022, https://doi.org/10.3390/s22155699).

---

## Author Comment (AC3)

I found the "Scientific Significance", "Scientific Quality" and "Presentation Quality"; Good.

The paper addresses relevant scientific and/or technical questions within the scope of NHESS, including new data and results which are presented in international standards.

The scientific methods and assumptions are valid and outlined clearly. The results are sufficient to support the interpretations and the conclusions. The title of the manuscript clearly and unambiguously reflect the paper's contents. The abstract provides a concise, complete and unambiguous summary of the work done and the results obtained. The title and the abstract are pertinent and easy to understand for a wide and diversified audience. The size, quality and readability of the figures are adequate for the type and quantity of data presented. Authors give proper credit to previous related work, and they indicate their contribution, clearly. The overall presentation is well structured, clear and easy to understand by a wide and general audience. The length of the paper is quite adequate; not too long, not too short.

The fluency of the paper is good, in general. I noticed some ambiguity in some of the statements which can be edited quickly and easily. You can kindly find my comments/corrections/suggestions which will improve and strengthen the submitted paper, I believe. I congratulate the writers and wish them success in their future work. Last but not least understanding the background noise levels and seismic network standards is very important for studies in earthquake locations and early warning systems. I also want to thank the journal editor for evaluating this submission which will be quite beneficial and enlightening for the data users of this seismic network.

Seismic data quality affects the earthquake monitoring capabilities of a seismic network significantly. More than that near-source strong motion records have great advantages for real-time estimation of earthquake magnitude by providing unsaturated recordings of moderate to large earthquakes to get fast and robust earthquake location and magnitude estimates. So, the quality of data from individual stations has to be estimated, evaluated and investigated constantly and regularly. In this framework, I found the submitted work of Fornasari et al. very valuable and precious. The results of the paper will be quite beneficial for the researchers who will use the data of the Italian Strong Motion Network. It is a necessity for each network to have similar studies investigating noise levels. So, I appreciate and encourage the submitted work and effort. I found this study, to have such an effort for Italian strong motion data, quite valuable. I would be in favour of the publication of this paper.

1. Line 1: "... with more than 700 stations." ITACA ($https : //itaca.mi.ingv.it/ItacaNet_32//station/search$) refers to 836 stations for Italian Strong Motion Network, FDSN presents (https://www.fdsn.org/networks/detail/I 523 stations. May you please provide a specific number, with a source?
   We update the information related with the stations so that it is easier to understand the information related with stations. The RAN consists of more than 700 stations of which 532 provided continuous data in the time range that we are interested in. Relatively to the number of stations reported by ITACA, it also considers temporary networks and uninstalled stations. We have the exact current number of installed stations as we perform real-time monitoring using their data but since this number changes constantly due to the addition and removal of stations we prefer not to provide a specific number.

2. Line 9: "... we focused on relatively short periods (5 s), interested by in anthropic noises." Line 9: You refer **"anthropic"** noise/activities/sources in Line 9, Line 24, Line 98, Line 119, Line 186, Figure 11 caption.
   You refer **"anthropogenic"** noises/sources in Line 5, Line 42, Line 75, Line 107, Line 165, Line 223
   You refer **"cultural"** noise in Line 15, Line 140, Line 159.
   If you are using these terms in the same meaning, please be more consistent in the usage. If you are referring to different meanings, please describe this.
   We decided to use the term "anthropogenic" to identify human-related noise, except in several occasions we decided to use "human" instead of "antropic".

3. Line 12: "... stations are located in densely populated areas such as the center of Naples, ...". ✓

4. Line 14: "noise levels dropped to 6.5 decibels in the daytime and 12.5 decibels on weekdays. ✓

5. Line 20: "On the other hand, noise can be used for the characterization of layers of the earth (Shapiro et al., 2005), moon (Larose et al., 2005) or Mars (Schimmel at al., 2021). "
   Please refer also to those additional sources:
   https://agupubs.onlinelibrary.wiley.com/doi/10.1029/2005GL023518 ✓
   https://agupubs.onlinelibrary.wiley.com/doi/epdf/10.1029/2021EA001755 ✓

6. Line 21-26: Please cite also (Stutzmann et al., 2000).
   (https://pubs.geoscienceworld.org/ssa/bssa/article/90/3/690/102808/GEOSCOPE-Station-NoiseLevels ) ✓

7. Line 48: " that has have been carried out for more than 25 years."
   We rephrased the sentence.

8. Line 64-Line67: Please cite DOI's, for these networks properly:
   Italian Strong Motion Network - 10.7914/SN/IT - ✓
   Friuli Venezia Giulia Accelerometric Network - 10.7914/SN/RF ✓

Irpinia Seismic Network - ?
We cited the paper dedicated to ISNet (`https://doi.org/10.1785/gssrl.78.6.622`) as ISNET do not have any registered DOI.

9. Line 75: Regarding your commentabout the change of the stations on Line 51-Line52; I may suggest you to perform 2022-year noise as a whole, in order to present a full understanding of the background noise for the mentioned networks. Then, to compare the results with the Covid-19 lockdown period provide more meaningful and stable results.
Currently, we perform our real-time monitoring only by using the servers at the department of civil protection without holding the seismic waveforms in our servers in Trieste due to a problem in our hard drives. Because of that, we are not able to do further analysis with the data later than April 2022. If we solve the space problem in our servers, we will add the rest of the 2022, if not, unfortunately, we cannot do it.

10. Line 112: "Numerous stations exceed the levels defined by Cauzzi and Clinton (2013)." May you please comment/discuss on the possible reasons of this situation.
In Table S1, number of stations that exceed the AHNM are provided. In periods between 0.04 second to 2 second there are numerous stations exceeds the AHNM. In Figure 1, we provided the noise sources and between 0.04 to 2 seconds culture noises are dominant in the seismic traces. Since the idea behind the integrated RAN network is to monitor the ground motions during an earthquake in the urban areas, many of our stations are positioned in the settlements. This may amplify the cultural noises and considerable number of stations exceed the upper noise limits.

11. Line 126: Diurnal and seasonal variations of seismic noise is well documented many years before 2021. You should also check (Stutzmann et al., 2000; McNamara and Buland, 2004), and please cite (Stutzmann et al., 2000), accordingly.
(https://pubs.geoscienceworld.org/ssa/bssa/article/90/3/690/102808/GEOSCOPE-Station-NoiseLevels ) ✓

12. Figure 1: Please mention what is "EM noise" and insert Cauzzi et al. (2014) to the References.
We change the caption to explicitly address the electromagnetic (previously EM) noise and also fixed the typo in Cauzzi and Clinton (2013).

13. Figure 4: Does this caption refers to Supplement S1 or Table 4? Please, edit given period band in the caption accordingly. " Median vertical component noise maps in one-third octave bands around a-g) 0.1 s, 0.25 s, 0.5 s, 1 s, 2 s, 5 s, 16 s, 32 s, and 80.6 s."
In Figure 4, the wrong figure was presented by mistake. We changed the figure and it can be seen in the updated version of the paper. We also put the same figure below

[Figure]

Figure 1: Median vertical component noise maps in one-third octave bands around a-g) 0.1 s, 0.25 s, 0.5 s, 1 s, 2 s, 5 s, 16 s, 32 s, and 80.6 s. Upper and lower limits of the color bar are defined by the model developed by Cauzzi and Clinton (2013). Vertical components are presented in the following figures and Electronic Supplement. Background noise levels of all calculated periods can be found in Figure S1.

---

## Author Comment (AC4)

This study is about the computation of the variations in ambient noise levels of the big Italian Strong Motion Network to evaluate the performance of the stations. The study is definitely important and useful for future studies. I appreciate the authors used a large dataset to establish the study by considering different periods and taking the advantage of COVID-19 lockdown period. The methods they used is a well-known and suited to the paper. The manuscript is written in good English. Although I like the idea of the paper which provides an excellent opportunity to exploit this large dataset for different time periods, I believe the paper still needs some significant revision. Please see my comments below.

**General Comments:**

1. The paper sometimes is lacking in quantification for the validation of the results appropriately, especially in the result section. For instance, I am a bit surprised there is no quantitative comparison with the results from the other networks around the world to evaluate the performance of the Italian Network in the Discussion part. I would definitely add one paragraph to the Introduction part, showing the previous studies and their ambient noise level with numbers, and compare&discuss them in the Discussion section.

   We have added numerical results related to the background noise levels of the network. Network-wise comparison would can be done but background noise information is highly dependent on the local conditions, especially in short periods. However, we use the High- and Low-Noise Model developed by Cauzzi and Clinton (2013) and use it as a baseline. There are other studies that provided background noise models for broadband seismic networks such as Peterson (1993) and D' Alessandro et al. (2021). However, broadband seismic stations have different sensibilities in different periods. Results in Figure 3-5,13, and 15 and Table 3. We realized that there is a problem in Figure 4 and replace it with another figure. You can see it below:

[Figure]

Median vertical component noise maps in one-third octave bands around a-g) 0.1 s, 0.25 s, 0.5 s, 1 s, 2 s, 5 s, 16 s, 32 s, and 80.6 s. Upper and lower limits of the color bar are defined by the model developed by Cauzzi and Clinton 2013. Vertical components are presented in the following figures and Electronic Supplement. Background noise levels of all calculated periods can be found in Figure S1.

2. The paper also needs additions and extra explanations because some details are missing (e.g., in the Method section).
   We improved the Method section of the paper

3. The organization of the paper is not well structured. The aim of the study is not given clearly. The sections sometimes don't show their actual points. While the Results section is very smooth, the Discussion part contains mostly the results of the study. Furthermore, the text sometimes contains repetitive sentences specifically in the Result, Discussion, and Conclusion parts. This does not make the article fully comprehensible.
   We added the aim of the study to Introduction section. We agree with the reviewer about the relatively weak Result section. To improve it, we quantify some of our results and presented them in Results section. We also reduce the repetitive parts as much as possible.

4. For the figures, the figure axis fonts are quite small and not readable. On the other hand, the authors can do zoom-in maps based on different coordinates instead of showing the whole land of Italy which makes the figures more catchy.

We increase the font sizes in several figures to increase the readability. In the early stages of the paper we had various selected stations to explain some features of the noise levels in specific periods. However, it increased the length of the paper and we believe that the information retrieved from those site-specific features do not increase the quality of paper. We also tried to zoom in the before-mentioned parts in figures but it makes the figures even more complicated. We believe providing the overall picture of the network is a better approach on visualizing the noise levels of the Italian territory.

**Minor comments:**

**Abstract**

1. Line 5: ...anthropogenic ...Please be consistent to write this term in the same way throughout the text also figure captions. For example, it is written as "anthropic" in Line 98.
We decided to use terms "anthropogenic" and "human activity" in general. In the updated version of the paper "anthropic" is used only in Figure 11 in which the term is used by the previous study.

**Introduction**

1. Line 49: ...RAN accelerometric network... Only "RAN" is enough here. Please be consistent with the abbreviations throughout the paper. ✓

2. Line 56: ...covid lockdown... Please be consistent (e.g. line 57: COVID-19) ✓

3. Line 62: ...more than 700 stations... How many exactly?
We update the information related with the stations so that it is easier to understand the information related with stations. The RAN consists of more than 700 stations of which 532 provided continuous data in the time range that we are interested in. We have the exact current number of installed stations as we perform real-time monitoring using their data but since this number changes constantly due to the addition and removal of stations we prefer not to provide a specific number.

4. Can authors add also a paragraph from previous studies and their findings that use the same or different methods? Please also describe clearly the aim and purpose of your study which is missing in the text.
We mention the previous studies in lines between 29-41. Model developed by Cauzzi and Clinton (2013) are used as a baseline and in many parts of the paper (eg. Figure 5, Table 3) comparisons can be found. In the discussion part some of our findings are compared with D'Alessandro et al. (2021)'s study since they both cover the same region.

**Data**

1. Please add a diagram to clearly show the sub-networks that are involved in National Accelerometric Network (A supplementary figure is fine).
The integrated RAN network is the combination of the following networks; i) the Friuli Venezia Giulia Accelerometric Network (RAF, Rete Accelerometrica Friuli Venezia Giulia in Italian, Costa et al. 2010) in the North-East Italy, owned and managed by the University of Trieste (UniTS) ii) Irpinia Seismic Network (ISNet, Weber et al. 2007) in the South of Italy, owned and managed by Analysis and Monitoring of Environmental Risk society (AMRA).
We updated Figure 2 and it can be seen below. In the updated version diamonds, stars, and circles represent RAN (IT), ISNet (IX), and RAF (RF) networks, respectively. Colorbar presents the completeness of the data. diamonds represent the RAN network regardless of the completeness values. In the zoomed frames, ISNet and RAF networks are given different colours (star for IX and circle for RF) to explain where the stations are located.

[Figure]

Data availability of the stations in a) 2019, b) lockdown period, and c) 2022. The close up boxes in lower left and upper right highlight ISNet (IX) and RAF (RF), respectively. Basemap data are retrieved from © Stamen Design.

We also created a Venn diagram for the networks. We believe in the updated Figure it is clear to see the relation between networks so we believe it is not necessary to put the diagram that we provide here to put also to the supplementary material.

[Figure]

Components of the integrated RAN network.

2. Could authors give more information about the stations (e.g., type of instruments, sensor, the cut-off frequency) used in their analysis? We added information related to instruments. The cut-off frequency is 80 % of Nyquist frequency of the station. To provide a snapshot of the status of the network for each of the three periods considered, we prepared the table below considering the first day of each period (the same table is added to supplementary material):

Table 1: Evolution of the sensors at integrated RAN stations.

| Sensors[a] | 2019[b] | 2020[c] | 2022[b] | Sampling rate [Hz] |
|---|---|---|---|---|
| Kinemetrics EpiSensor | 177 | 274 | 370 | 200 |
| Syscom ms2007 | 3 | 3 | 87 | 200 |
| Kinemetrics FBA-23 | 23 | 27 | 35 | 200 |
| Guralp CMG-5T | 0 | 15 | 20 | 125 |
| Reftek 147A | 0 | 0 | 4 | 200 |
| CFX US4H | 1 | 1 | 2 | 200 |
| Lunitek FB | 1 | 1 | 1 | 250 |

[a] Equipped with 24bit recorders

[b] Status at January 1st

[c] Status at March 9th

**Results**

1. Line 105: ...0.1 s, 0.25 s, 0.5 s, 1 s, 2 s, and 5 s...Why did the authors choose these periods? Did they do any analysis to determine them? A spectrogram plot would be helpful for selected stations in order to understand dominant periods. Please explain it in more detail in the Data&Method section, not in the Result section.
   We believe that the anthropogenic sources are the main noise sources in our network. Hence we selected several periods in which anthropogenic sources are dominant (see Figure 1). 2 s and 5 s are chosen since there is information related to wind and sea can be found. Although the selection of the periods is arbitrary, we considered the choices made in previous studies combined with our interest to see the noise levels in our network. We would like to see if the strong motion stations are susceptible to these noise sources. We have not plotted any spectrogram but we provide multi-year noise level changes in various stations. As the reviewer mentioned, the data can be visualized in many ways. For the sake of simplicity, we use Figure 3 to show how in different periods noise levels are changing.

2. Line 125: ...The results show that winters are noisier than the summers...Can authors quantify how much (with numbers)?
   As told in the line 125, in winter longer periods are noisier than the summer time. In 5 s noise level differences are not really in an agreement with 87 noisier stations with respect to 49 quieter stations. In longer periods, 8 s, 16 s, and 32 s, number of stations that are noisier in winter with respect to summer are 117, 121, and 115. Median dB changes for those 4 periods are 0.56 dB, 0.99 dB, 1.55 dB, and 1.33 dB, respectively. We add this information to the paper.

3. Line 129 – 138. I don't see any indication of results in these two paragraphs. Can you specify the ambient noise levels for the different periods that you selected for the lockdown period? Authors should mention the result in the text as they plotted in the figures in this section. If the authors don't mention any results, the first paragraph rather seems related to the Introduction part while the second one is data and/or method related.
   We added provided overall results related to the lockdown period to Section 4.1.

**Discussion**

1. Please see my general comments above. ✓

2. Line 190-194. Did the authors compare their results with the studies mentioned in this paragraph during the lockdown period?
   We add more detailed information related to several previous studies about the topic. However, depending on the approach to the covid-19 effects, in the previous studies different approaches have been used to see the effects. Hence, it is not always possible to do a direct comparison.

**Conclusions**

1. Please avoid redundant sentences here and emphasize the main findings, contribution, and significance of your work. ✓

**Figures**

1. Figure 1: I am not sure if this figure should be included in the main text. It can go to the Supp. Material. Please see my general comments about the figures above.
   We believe that Figure 1 provides a nice characterization of the noise in seismic records. It may help readers to interpret the noise level that we present in this study. Hence, we believe that Figure 1 should be in the main text. Figure 1 has also been improved by adding the one-third-octave bands used in our analysis to show the expected contributions to the recorded noise.

Lastly, I believe the paper could be improved in many ways with some additions and restructuring as suggested above and published in the journal NHESS after applying the required revision.

---

## Referee Report (RR1)

– L.23 Earth merits to be capitalised as the other celestial bodies.
– L.30-31 "To be able to monitor the seismic sources, seismic networks require knowledge about the noise content of the networks", This sentence is not necessarely true. One could detect events by observing P-picks without knowing anything about the noise level and their sources. This counts when one is interested in more sophisticated analysis ad, for example, determining the detection threshold.
– L61-66, This part could be shortened with a clear statement that mention that the lockdown period can be used to determine the anthropogenic component of the background noise.
– L.73-75, It remains unclear to me why data from 2022 are not included in this study. The authors in an early answer wrote that they were evaluating it. The authors claim they provide a better coverage.
– L.100 studied by grouping", better "grouped"?
– L.108, "20 randomly selected", I do not understand the motivation for a random selection, and the authors forget to explain it. If the intention was to present the variety of noise levels, how can the reader interpret the differences without knowing the different location, soil etc?  At least the 20 should be marked on map. I would suggest to make an arbitrary selection in which different soils and different type of urbanisation are represented. I do not understand why the authors only represent six narrow beans and not the full PSD. The authors do not explain why the "periods of interest" are the 6 reported at line 109. This is somehow in contrast this sentence at lime 112 "we are mainly interested in periods less than 5s". Formally speaking, the latter selected bean (5s) is out of this range of interest. The above sentence also contrast with the definition given in line 55 in which 5s is included.
– L124+ Noise decreases over night, ok. But from figure 6, I see large patches of white markers as in Tuscany in which there is not such a decrease. This is not mentioned neither discussed. As mentioned above this would be the key aspects that make this paper valuable for publication.
– L127-129 we have 5 periods, 5 median values and 6 number of noiser stations. Can't be.
– L.163, "italian strong motion network" is something different from RAN or Integrated RAN? This was never defined although it is mentioned in the title, here and in the caption of figure S1.
– L.178 and following, The discussion in this paragraph is not exhaustive. In frame d) we have large patterns indicating "no variation" while in frame f) the Pianura Padana is dominated by blue. This cannot be neglected. These are, in my opinion the key aspects that would make this manuscript valuable.
– L.181-184, this is a clear sample of my general comment. We are in the section "Discussion" dedicated to the discussion of the results and the section mention previous results and give motivation for the results of this

paper. But this is not enough, how can I be certain that we are observing wind or sea or whatever else if the authors do not show it. One could compare noise variation with wind speed, or with sea storms, or traffic data. This is in my opinion an incorrect approach to data analysis.

– L.193, I am confused, Noiser or quiter? Where can I see this?

– L.226-232, What is the added value of this study with respect to what observed by Poli et al, or by Piccinini et al? The latter also discusses spatial patters and economical motivations. It is not enough to write that noise generated by human circulation decreased when people where locked down. This is not a novel discovery.....

– L.233-236, This are "results" not discussion of them.

– L255 and 244, for those not familiar with the area, DTS2 appears located in two different places.

– All toponyms should be marked on maps, authors cannot presume that the reader knows where Ischia, or Naples or Palata are located on maps.

– Section 5.3, This section presents some results, and it does not include a discussion of them in the contest of the paper.

– As remarked in my previous review, accuracy is crucial when writing and when reporting information. The coordinates of Palata differ in section 5.3 and in figure 15. Moreover 41.886, when truncated to a two digit number is 41.89.

– L274 "have higher noise levels than the AHNM" should be "have noise levels higher than AHNM"?

– L278, the example of CSA7 is confusing, the station was never mentioned before (except in one figure of the supplementary material). I think that CSA7 should be explicitly inserted in the discussion or results section before citing it in the conclusion. How can the reader understand this example?

– L278 "some of these stations" some is vague. I think the author could evaluated the number of percent of station located in towns.

– L279 "the true nature of the ground motion if there is a strong earthquake nearby" this is a generic and vague statement. Data and analysis could be used to provide a quantitative result. How many can record the full waveform of a magnitude 3 or 4 or 2.5 with a proper Signal–to–Noise ratio?

– L280 "capabilities of the stations" is a vague concept.

– L281, again, "The surrounding conditions for RAN stations within settlements are variable and have noticeable effects on the noise levels" is this a result of this study? How the authors distinguished the different condition for stations within settlements? How can we get to this conclusion?

– L289, Why this is observed at some stations and not at others? Instrumental difference, site difference?

– L295 The author touch the fact the accelerometers are "deaf" and, in absence of strong ground motion they record the self noise of the instrument. Should not this pointed out at the beginning to restrict the detection capability of the instrumentation used instead of using it as an

empirical conclusion?

- L298-299 "an average reduction in the noise level of 1.0 dB (and up to 2.9 dB at 0.0625 s) during the daytime". Neither in the manuscript nor in the supplementary material this numbers (1.0, 2.9 and 0.0625) can be found by myself
- Figure 1, D'Alessandro et al is 2020 in the figure and 2021 in the caption.
- Figure 4. Again on the care of details, The figure as nine frames labelled from a) to l). In the caption it is mentioned a-g). Moreover the latter is for 80.6s but this period is never discussed in the manuscript. "Vertical components are presented in the following figures and Electronic Supplement." This is not clear.
- Figure 5, the noise model are NLNM and NHNM by Peterson or A... by Cauzzi and Clinton, the caption is confusing.
- Figure 6, The caption is not correct. I suspect the figure represents the different for each station and for each period between the median of the noise level at daytime and nighttime.
- Figure 14, I fear that the authors while using the image from Google did not follow the reproduction rules set by NHESS and by google. please check.
- Table 3, caption. Stations or number of stations? "with higher" or "with noise level higher"
- Table S1, To me "evolution of the sensors" means how each sensor evolved/changed". Looking at numbers, I suppose the authors are referring to the change over time of the number of sensors divided by type.
- Figure S1, The authors did not describe what is the difference between the single maps or, in other words, what the number on top of each frame is. Moreover, it is a common practice to label each frame with a letter or a number.
- Figure S2, il the title, "Difference" should not be capitalized. In the title the author us the dash without spaces (Weekday-Weekend) while in the caption the use it with spaces (2019 - 2022). Should this follow the same rule?
- Figure S2 caption, punctuation is messy
- Figure S3, I suggest to include also M5 and M6 in this figure. And to carefully discuss it. How many station would miss to record correctly the full waveform for the ground shaking of a M5 or M4? Moreover, different lines should be described in the caption.
- Table S3, caption. The authors miss to mention what is higher than AHNM, I suppose they are referring to "Stations with noise level higher then AHNM".
- Table S3. Period and AHNM require the measure unit besides them. No of station should be no of stations.

---

## Author Response (AR2)

First of all, we thank the Reviewer for the insightful comments. We used this second round of revisions trying to further improve the quality of the manuscript.

Following the suggestion of Reviewer #2, and as requested by multiple Reviewers during the first revision, we decided to add the analysis of the complete 2022 dataset: such analysis was not originally presented as the dataset was not available at the moment of the first submission. In light of this, we valued unnecessarily the analysis of 2019 data, at first introduced to study the "long-term" variations of noise levels: due to the recent changes in the network, the data from 2019 could only provide a partial coverage of the network.

Based on the whole 2022 dataset, we also developed the Italian Accelerometric Noise Models: these should provide more exhaustive results and can be of interest to future projects.

The spatial and temporal variability has been studied with more emphasis but, while some patterns emerge from our analysis on a national scale, more localized studies are required to investigate the causative nature of the noise at specific locations.

Considering the mentioned changes to the manuscript, we decided to remove the sections related to noise variations during COVID-19 lockdown and traffic noise: while these analyses can provide some insight into the characterization of anthropogenic noise, their contribution to the manuscript (in its actual stage) is minor.

In the following, the relevant changes proposed during the second revision of the manuscript are listed:

- overall changes to the structure of the manuscript to improve its readability;

- removed the COVID-19 lockdown analysis;

- removed traffic noise case study;

- added the analysis of (and limited to) the complete 2022 dataset;

- introduced the Italian Accelerometric Noise models;

- added quantitative comparison with D'Alessandro et al. (2021) models;

- improved spatial and temporal analysis of noise levels;

- expanded Discussion section.

It follows the Reviewers' comments with our detailed replies (in blue): straightforward comments with which we agree are just marked with a green tick (✓).

**Reply to Reviewer 1**

Dear authors, the manuscript is surely improved from the first run. Honestly, I would have chosen another organization of the work but if the other reviewers like it, there is no further problem also for me, and the manuscript is acceptable for the publication after (very) minor revision. In the following my suggestions...
I think that an English native language reader could improve the text. If not possible consider at least that in the Introduction section there is an overuse of the adjective "seismic". Many "seismic" can be deleted or replaced with other synonyms (e.g. seismic events=earthquakes, seismic noise=ambient noise and so on...).

1. Line 23: seismic recorder should be plural (and "seismic" could be deleted) ✓

2. Section 2 (line 68-70): It should be obvious but I add something to say clearly that data are acceleration and using what unit of measure (cm/s$^2$, g) ? This information is useful also to understand the method, the PSD computation
   We added this information into the Method section as below
   "Considering only the vertical components at the stations, each daily recording in acceleration is divided into . . . "
   Raw data are stored in counts.

3. Line 94: "the instrument response is then removed from the PSD" I don't understand the meaning. Do you mean that averaging limits the electronic noise produced by the instruments?
   After the calculation of PSD of stations, it is divided by the instrument response of each station period-wise.

4. Line 165: a space between 532 and stations is required. ✓

Good luck

**Reply to Reviewer 2**

The manuscript is relevant because it evaluate the noise level of the Italian Accelerometric network. Among the discussion about the sources of noise and their variability over time, this is the baseline for determining the detection capability of the network in terms of magnitude and distance. In my view, the manuscript remains not mature for publication and I suggest to consider a major revision. This stems from two main points:

1. there is still confusion between results, discussion and observation from previous studies.

2. there is not an exhaustive discussion of the results. Moreover, the author propose reasonable motivation for some of the observed features but did on explore them.

My suggestion is to restructure the manuscript by:

1. removing the section dedicated to covid and to car passages.
   We decided to accept the suggestion about removing covid and car passages and the suggestion about adding 2022 data. In the end, we limited the analysis only to the 2022 data and its spatial features. To enrich the paper, we decided to insert the background noise model for Italian strong motion data which was originally part of another article we would like to publish. As a result of this decision, some of the comments/suggestions/questions became irrelevant.

2. strongly focus on the noise level spatial variability over time and space
   We believe we provided more information about the spatial and temporal variability of the background noise in the revised version. However, there are various results without or with limited explanation such as noisier weekends in longer periods. Since strong motion stations are not very susceptible in the long period effects, we do not provide any detailed explanations to these results.

3. discuss the motivation for the spatial variability by considering the soil difference at the deployment sites, the urban/countryside distinction, the presence of industrial sites.
   We looked at the local soil classes but there is not much of a variation with the soil classes. Furthermore, we compared the noise level with the geological formation. Furthermore, we tried to link the noisy stations with geological settings but there is no clear correlation between geological class and noise levels. Industrial sites may play a major role on the noise levels but we decided to use only the land usage types instead of characterizing more than 500 stations' surroundings manually. In fact, there is a clear example of daily noise level change in DMN station that is located next to a hydroelectric power plant (Figure 1). One can follow the working shifts of the facility quite easily. However, in other stations there is no such clear pattern. Hence, it is hard to justify the noise levels just by industrial sites. This is why we focused more on settlements, in general.

[Figure]

Figure 1: Daily noise variation of DMN station (lat:44.315 lon:7.271).

Referee Report:

1. L.23 Earth merits to be capitalised as the other celestial bodies. ✓

2. L.30-31 "To be able to monitor the seismic sources, seismic networks require knowledge about the noise content of the networks", This sentence is not necessarily true. One could detect events by observing P-picks without knowing anything about the noise level and their sources. This counts when one is interested in a more sophisticated analysis ad, for example, determining the detection threshold.
   Observing P wave may be an easy task in large earthquakes but in small earthquakes P wave arrival can be overshadowed by the noise. Understanding the anthropogenic sources may improve the precision of determining the epicenter of seismic events. We are using a data band-pass filtering routine for earthquake detection with dynamically selected upper and lower bands. The bands depend on the background noise of the station. Hence, for our network, background noises affect the earthquake detection capabilities. Furthermore, the event detection routine that we are currently using in our seismic monitoring task miscalculates the epicenter of an event if there is a car passing near a station at a far away distance from the seismic event when seismic signals reach the stations at the same time. This is actually why we added the vehicle noise section to the paper in the first place.

3. L61-66, This part could be shortened with a clear statement that mention that the lockdown period can be used to determine the anthropogenic component of the background noise.
   We deleted the sections related with the covid. Hence this paragraph is not a part of the paper anymore.

4. L.73-75, It remains unclear to me why data from 2022 are not included in this study. The authors in an early answer wrote that they were evaluating it. The authors claim they provide a better coverage.
   2022 was not available due to storage problems in our servers that have been solved during the discussion process of the paper. As mentioned above 2022 is the only data source of the paper now.

5. L.100 studied by grouping", better "grouped"? ✓

6. L.108, "20 randomly selected", I do not understand the motivation for a random selection, and the authors forget to explain it. If the intention was to present the variety of noise levels, how can the reader interpret the differences without knowing the different location, soil etc? At least the 20 should be marked on map. I would suggest to make an arbitrary selection in which different soils and different type of urbanisation are represented. I do not understand why the authors only represent six narrow beans and not the full PSD. The authors do not explain why the "periods of interest" are the 6 reported

at line 109. This is somehow in contrast this sentence at lime 112 "we are mainly interested in periods less than 5s". Formally speaking, the latter selected bean (5s) is out of this range of interest. The above sentence also contrast with the definition given in line 55 in which 5s is included.

20 randomly selected stations were inserted into the paper for data visualization purposes (a similar example can be seen in Figure 2 of Lecocq et al. (2020, doi: https://doi.org/10.1126/science.abd2438)). Since it has no significance apart from the data visualization we removed the image from the paper. We added a new figure (Figure 5) which shows the full PSD of several stations with different land usage types. In terms of periods of interest, we decided to keep the 5s and add the missing information to the necessary parts of the text.

7. L124+ Noise decreases over night, ok. But from figure 6, I see large patches of white markers as in Tuscany in which there is not such a decrease. This is not mentioned neither discussed. As mentioned above this would be the key aspects that make this paper valuable for publication.

We discuss these stations in lines 186+ and 201+ in the Discussion section.

8. L127-129 we have 5 periods, 5 median values and 6 number of noiser stations. Can't be.

We added the 5s to the periods and number of periods and median values are matched.

9. L.163, "italian strong motion network" is something different from RAN or Integrated RAN? This was never defined although it is mentioned in the title, here and in the caption of figure S1.

In Line 163 it refers to integrated RAN. Both in Line 163 and in the caption of Figure S1 are replaced with the RAN. Venn diagram of the Italian Strong Motion network can be seen in the figure below. The entire structure (integrated RAN) defines the Italian Strong Motion network. As explained in L45, in the paper RAN refers to integrated RAN.

**Integrated RAN**

[Figure]

Components of the integrated RAN network.

10. L.178 and following, The discussion in this paragraph is not exhaustive. In frame d) we have large patterns indicating "no variation" while in frame f) the Pianura Padana is dominated by blue. This cannot be neglected. These are, in my opinion the key aspects that would make this manuscript valuable.

About the weekday-weekend difference in the previous version of the paper observations of the review. In the revised version in which only the 2022 data is taken into consideration, results are slightly different. We discuss the possible explanations of the results in Lines 207+.

11. L.181-184, this is a clear sample of my general comment. We are in the section "Discussion" dedicated to the discussion of the results and the section mention previous results and give motivation for the results of this paper. But this is not enough, how can I be certain that we are observing wind or sea or whatever else if the authors do not show it. One could compare noise variation with wind speed, or with sea storms, or traffic data. This is in my opinion an incorrect approach to data analysis.

We decided to delete this paragraph.

12. L.193, I am confused, Noiser or quiter? Where can I see this?

Weekdays are noisier than the weekends. We changed "weekends" with "weekdays".

13. L.226-232, What is the added value of this study with respect to what observed by Poli et al, or by Piccinini et al? The latter also discusses spatial patters and economical motivations. It is not enough to write that noise generated by human circulation decreased when people where locked down. This is not a novel discovery. . . .

As mentioned before covid related parts of the paper are erased. Hence this part is not a concern anymore.

14. L.233-236, This are "results" not discussion of them.
    As mentioned before covid related parts of the paper are erased. Hence this part is not a concern anymore.

15. L255 and 244, for those not familiar with the area, DTS2 appears located in two different places.
    In Line 238, we provide the information about the building where DST2 is located. However, at Line 228 we provide the geological settings of the station.

16. All toponyms should be marked on maps, authors cannot presume that the reader knows where Ischia, or Naples or Palata are located on maps.
    We added the cities mentioned in the paper into the maps. We added a small map inside Figure 10 to show the location of Trieste on map. Locations of the stations presented in Figure 5 are not provided on the map since the significant information of the stations are their land usage not their location on map.

17. Section 5.3, This section presents some results, and it does not include a discussion of them in the contest of the paper.
    Sections related to vehicle noise are deleted.

18. As remarked in my previous review, accuracy is crucial when writing and when reporting information. The coordinates of Palata differ in section 5.3 and in figure 15. Moreover 41.886, when truncated to a two digit number is 41.89.
    We deleted the coordinates provided in Figure 14 since they are already given in Section 5.3.

19. L274 "have higher noise levels than the AHNM" should be "have noise levels higher than AHNM"? ✓

20. L278, the example of CSA7 is confusing, the station was never mentioned before (except in one figure of the supplementary material). I think that CSA7 should be explicitly inserted in the discussion or results section before citing it in the conclusion. How can the reader understand this example?
    We deleted the example of CSA7.

21. L278 "some of these stations" some is vague. I think the author could evaluated the number of percent of station located in towns.
    We updated Table 5 by adding the land usage distribution of the stations that exceed the AHNM and referring it in L243.

22. L279 "the true nature of the ground motion if there is a strong earthquake nearby" this is a generic and vague statement. Data and analysis could be used to provide a quantitative result. How many can record the full waveform of a magnitude 3 or 4 or 2.5 with a proper Signal-to-Noise ratio?
    We agree with the vagueness of the sentence. We added a paragraph to discussion and added Brune's corner frequency values to Figure 4 to show the capabilities of the network. We also provided an earthquake example in supplementary material to show a real case example of the effect of background noise.

23. L280 "capabilities of the stations" is a vague concept.
    We rephrase the sentence as follows "Depending on the nature of the future station installations and studies, noise levels of RAN (Figure 4) may give an insight into the suitable locations for the deployment.".

24. L281, again, "The surrounding conditions for RAN stations within settlements are variable and have noticeable effects on the noise levels" is this a result of this study? How the authors distinguished the different condition for stations within settlements? How can we get to this conclusion?
    Analysis of the background noise levels of DST2 and CARC stations can be an example on this. They both located in settlements with different background noise level.

25. L289, Why this is observed at some stations and not at others? Instrumental difference, site difference?
    Day-night variations reduce significantly in the longer periods. If we plot all the subfigures in Figure 7, in longer periods we would get almost no daily variations. Furthermore, in the longer periods our stations are 'deaf' hence the interpretation of the daily variation of these periods are tricky. We deleted the sentence and add another conclusive sentence related with day-night variance: "The difference is relatively low in the stations located on the mountainous parts of North-East Italy." and in Discussion we add: "In North-East Italy, there are several station with relatively low daytime-nighttime difference. These stations are located far away from all settlements and located on mountainous parts of Italy.".

26. L295 The author touch the fact the accelerometers are "deaf" and, in absence of strong ground motion they record the self noise of the instrument. Should not this pointed out at the beginning to restrict the detection capability of the instrumentation used instead of using it as an empirical conclusion?
    It can be expected to not see the long period effects in strong motion stations. However, there are not so many studies about this topic. Cauzzi and Clinton (2014) mentioned briefly that there are not significant variations in long periods but previous studies do not provide any specific analysis. In this paper, we show that in long periods accelerometric networks do not provide and seasonal variations. So

reader can compare the long period results, for example Antony et al. (2022), and see the differences between broadband and strong motion records in long periods.

27. L298-299 "an average reduction in the noise level of 1.0 dB (and up to 2.9 dB at 0.0625 s) during the daytime". Neither in the manuscript nor in the supplementary material this numbers (1.0, 2.9 and 0.0625) can be found by myself
As mentioned before covid related parts of the paper are erased. Hence this part is not a concern anymore.

28. Figure 1, D'Alessandro et al is 2020 in the figure and 2021 in the caption. ✓

29. Figure 4. Again on the care of details, The figure as nine frames labelled from a) to I). In the caption it is mentioned a-g). Moreover the latter is for 80.6s but this period is never discussed in the manuscript. "Vertical components are presented in the following figures and Electronic Supplement." This is not clear.
We changed the annotations and now they cover all subfigures. We mentioned about the vertical components since we only use the vertical components and exclude the horizontal ones. However, to better explain this fact, we specified the selection of the vertical component at Line 59 of Data section and add a sentence to Line 83 in Method section.

30. Figure 5, the noise model are NLNM and NHNM by Peterson or A. . . by Cauzzi and Clinton, the caption is confusing.
Model acronyms are wrong and they correct ones are ALNM and AHNM defined by Cauzzi and Clinton.

31. Figure 6, The caption is not correct. I suspect the figure represents the different for each station and for each period between the median of the noise level at daytime and nighttime.
Reviewer is correct. In both Figure 6 and Figure 7 differences are presented between day and night and weekday and weekend.

32. Figure 14, I fear that the authors while using the image from Google did not follow the reproduction rules set by NHESS and by google. please check.
Source of the satellite data is added to the caption of the Figure.

33. Table 3, caption. Stations or number of stations? "with higher" or "with noise level higher". ✓

34. Table S1, To me "evolution of the sensors" means how each sensor evolved/changed". Looking at numbers, I suppose the authors are referring to the change over time of the number of sensors divided by type.
Since we only use 2022 data, this is irrelevant now. In Table S1, we provide the periodwise limits of Lower (IALNM) and Higher (IALNM) Limits of the Italian Accelerometric Noise Model.

35. Figure S1, The authors did not describe what is the difference between the single maps or, in other words, what the number on top of each frame is. Moreover, it is a common practice to label each frame with a letter or a number.
We updated the titles of the figures and now periods are specifically written as a title of each figure. Furthermore we labelled each subfigure with Roman letters.

36. Figure S2, il the title, "Difference" should not be capitalized. In the title the author us the dash without spaces (Weekday-Weekend) while in the caption the use it with spaces (2019 - 2022). Should this follow the same rule?
Figure S2 is changed after the revision. Now Figure S2 provides information about Carciotti Palace (formerly Figure 12).

37. Figure S2 caption, punctuation is messy
Figure S2 is changed after the revision. This figure does not exist anymore.

38. Figure S3, I suggest to include also M5 and M6 in this figure. And to carefully discuss it. How many station would miss to record correctly the full waveform for the ground shaking of a M5 or M4? Moreover, different lines should be described in the caption.
M5 and M6 are added to the Figure 4 in the manuscript also to Figure S3 in supplementary material. We also add the interpretation of the Figure S3 in the supplementary material.

39. Table S3, caption. The authors miss to mention what is higher than AHNM, I suppose they are referring to "Stations with noise level higher then AHNM".
Reviewer is correct (except it is Table S2). Caption of the table is updated as "Stations with noise level higher then AHNM in the network (median and 2.5 statistics).".

40. Table S3. Period and AHNM require the measure unit besides them. No of station should be no of stations.
Periods are in seconds. "(s)" is added next to "Period" in the first column.